# The surface mass balance and near-surface climate of the Antarctic ice sheet in RACMO2.4p1

Christiaan T. van Dalum[1,2], Willem Jan van de Berg[1], Michiel R. van den Broeke[1], and Maurice van Tiggelen[1]

[1]Institute for Marine and Atmospheric Research, Utrecht University, Utrecht, the Netherlands
[2]Royal Netherlands Meteorological Institute, De Bilt, the Netherlands

**Correspondence:** Christiaan T. van Dalum (christiaan.van.dalum@knmi.nl)

**Abstract.** This study presents a new near-surface climate and surface mass balance (SMB) product for Antarctica for the historical period (1960-2023) using the updated version of the Regional Atmospheric Climate Model (RACMO), version 2.4p1 (R24). We study the impact of the major updates implemented in R24 on the climate of Antarctica, and show that the SMB, surface energy budget, albedo, pressure, temperature and wind speed compare well with observations. Compared to preceding RACMO versions, the advection of snow hydrometeors and alterations in the blowing snow parameterization affect the SMB, resulting in more precipitation in the mountains of West Antarctica and the Antarctic Peninsula, and an alternating pattern of SMB changes in the coastal zones of East Antarctica. Integrated over the ice sheet, including ice shelves, the modeled SMB is 2546 Gt yr$^{-1}$, with an inter-annual variability of 133 Gt yr$^{-1}$. Melt fluxes are small in Antarctica, at 124 Gt yr$^{-1}$ and an inter-annual variability of 31 Gt yr$^{-1}$, but can be significant on ice shelves, locally exceeding 500 mm water equivalent yr$^{-1}$. The presence of melt water in snow compares remarkably well with remote sensing observations and has improved compared to the previous operational RACMO version, 2.3p2. Temperature, upward and downward shortwave radiative fluxes and albedo are modeled well compared to in-situ observations. The longwave downward radiative and turbulent fluxes, however, require further model developments.

## 1 Introduction

The Antarctic ice sheet (AIS) plays a significant role in the climate system (Fretwell et al., 2013; Noble et al., 2020; Brace-girdle et al., 2020; Schneider et al., 2012). The contemporary climate of the AIS has been relatively stable, but studies show that the AIS is losing mass in recent decades (Harig and Simons, 2015; Rignot et al., 2019; Shepherd et al., 2018). At the surface, virtually all of the AIS is characterized by a net mass gain, as runoff to the ocean is typically much lower than snow accumulation (Mottram et al., 2021; Van den Broeke et al., 2023). The surface mass balance gains follow snowfall variability, which can be considerable from year-to-year (Davison et al., 2023), and even reverse the recent mass loss in high accumulation years (Wang et al., 2024). Most of the annual mass loss occurs as calving of icebergs and submarine melting of ice shelves, the floating extensions of ice sheets that surround most of the AIS (Rignot et al., 2013; Greene et al., 2022). Surface processes, however, play a crucial role in ice-shelf instability, as atmospheric conditions through melt events lead to the formation of melt

water ponds, potentially resulting in hydrofracturing (Scambos et al., 2009; Lai et al., 2020; Van Wessem et al., 2023b). In recent decades, higher temperatures both at the ice-water interface and in the atmosphere have led to instability and collapse of ice shelves, like Larsen A and Larsen B, and more ice shelves are susceptible to warming (Glasser and Scambos, 2008; Gilbert and Kittel, 2021; Van Wessem et al., 2023b). A thinning or collapsed ice shelf may lead to decreased buttressing, resulting in higher ice velocities upstream and increased discharge (Royston and Gudmundsson, 2016; Haseloff and Sergienko, 2022), which is the root cause of recent net mass losses of the AIS (Otosaka et al., 2023). The net accumulation of snow, i.e., the surface mass balance (SMB), therefore plays a crucial role in the stability of ice shelves and the AIS (Shepherd et al., 2018).

The SMB of a glaciated surface is determined by the difference between accumulation and ablation. In Antarctica, accumulation primarily occurs as solid precipitation, but riming and accumulation due to blowing snow can be significant on a local scale (Gadde and Van de Berg, 2024). Ablation is typically the result of sublimation or drifting snow erosion (Richter et al., 2021; Gadde and Van de Berg, 2024). Melt can be significant around the margins of the AIS, but almost all melt refreezes locally and runoff is often negligible (Kingslake et al., 2017; Van Wessem et al., 2018; Jakobs et al., 2020). Compared to the Greenland ice sheet, domain-integrated melt is several factors smaller on the AIS (Van den Broeke et al., 2023). The climate of Antarctica is characterized by strong regional differences. The interior of East Antarctica, for example, is characterized by a dry and cold polar desert, where sublimation can have a significant impact on the SMB, while the SMB of the western side of the Antarctic Peninsula (AP) is dominated by extreme snowfall amounts (Picard et al., 2019; Turner et al., 1997). Consequently, observations with high spatial and temporal coverage are necessary to capture all spatial variability. The remoteness and extreme conditions of Antarctica, however, results in a scarcity of observations (Eisen et al., 2008). Satellite observations are valuable to fill in the gaps, but technical limitations, such as the inability of optical sensors sensitive for visible light to measure during cloudy conditions or during the long darkness of the Antarctic winter, often results in missing or faulty data (Stroeve et al., 2013; Souverijns et al., 2018). Furthermore, some processes cannot be measured remotely, in particular the partitioning of melt energy into surface energy balance components or detailed characteristics of the firn (Van den Broeke et al., 2023). Apart from their inherent uncertainties, atmospheric climate models are able to estimate these processes and provide ice-sheet wide information that cannot be provided by in situ or remote-sensing observations on a high temporal and spatial resolution (Noël et al., 2018; Agosta et al., 2019).

General circulation models simulate the climate of the entire globe, but are often limited by a coarse resolution (>20 km) and usually do not include specific routines to describe glaciated surfaces accurately. Regional climate models (RCMs), however, typically have a higher spatial resolution and can be dedicated to specific regions. One such model, the polar (p) version of the Regional Atmospheric Climate Model (RACMO), is used in this study. It is specialized in modeling glaciated surfaces by having a dedicated glaciated surface tile. For this tile, parameterizations are applied to accurately model snow and ice, such as a multi-layer snow scheme and densification and grain size evolution parameterizations (Ligtenberg et al., 2011; Ettema et al., 2010). In previous studies, RACMO has been used to study the climate and SMB of the Antarctic and Greenland ice sheets (Van Wessem et al., 2018; Noël et al., 2018), but also to study smaller regions like Iceland or Patagonia (Noël et al., 2022; Lenaerts et al., 2014). Furthermore, RACMO has been used to study processes like the albedo of snow (Van Dalum et al., 2020; Jakobs et al., 2020) and turbulence on ice sheets (Van Tiggelen et al., 2023).

Recently, a new model version, RACMO2.4p1 (R24, Van Dalum et al., 2024b), has been developed and includes an update of the European Centre for Medium-Range Weather Forecast (ECMWF) Integrated Forecast System (IFS) physics, which is embedded in RACMO, from cycle 33r1 to cycle 47r1 (ECMWF, 2020). This includes revisions in the turbulence, convection, surface, aerosol, cloud, radiation, lake and turbulence schemes. More revisions are added that are related to the cryosphere but are not part of the IFS cycle 47r1, such as a new fractional land-ice mask, updated blowing snow scheme (Gadde and Van de Berg, 2024) and a multi-layer snow scheme for seasonal snow on non-glaciated surfaces. Using R24, we present here a new near-surface climate and SMB product for the historical period (1960-2023) for Antarctica on an 11 km horizontal resolution grid. We evaluate the output by studying several processes and compare it with remote sensing and in-situ observations and with the previous RACMO version, RACMO2.3p2 (R23p2, Van Wessem et al., 2018).

This manuscript continues with an overview of RACMO and recent updates in Sect. 2.1, followed by definitions and descriptions of in situ and remote sensing observation data sets in Sect. 2.2 and Sect. 2.3. Section 3 shows results for the SMB, a comparison with the AntSMB data set and analysis of the SMB for all drainage basins. Melt and the presence of liquid water in snow are discussed and compared to observations in Sect. 4. The near-surface climate, including the temperature, surface energy balance and albedo are discussed in Sect. 5. Case studies about melt in 2005 and seasonal snow in the McMurdo Dry Valleys are analyzed in Sect. 6. Conclusions and final remarks are given in Sect. 7.

## 2  Methods and data

### 2.1  Regional climate model RACMO

The Regional Atmospheric Climate Model (RACMO) employed in this study is a hydrostatic model and consists of two major parts: the dynamical core of the HIRLAM model (Undén et al., 2002) and the physics parameterizations of the ECMWF's IFS (ECMWF, 2020). The HIRLAM dynamical core uses a semi-implicit and semi-Lagrangian scheme to determine the origin of moving air parcels within a Eulerian frame (McDonald and Haugen, 1992). The physics parameterizations of the IFS include processes relevant on the sub-grid scale, like turbulence, radiation and surface processes. At the lateral boundaries of the model domain, wind speed, temperature, pressure and humidity are prescribed from reanalysis or a GCM with multi-level data. Similarly, sea-ice extent and sea surface temperature are also prescribed at the sea-surface boundary.

The polar (p) version of RACMO, which is maintained and developed at the Institute for Marine and Atmospheric Research Utrecht (IMAU), includes a dedicated glaciated surface tile. For this tile, parameterizations have been developed for snow and ice, including a multi-layer snow scheme, dry and wet snow metamorphism, snow densification, blowing snow, melt formation, melt percolation and snow albedo. Recently, a new model version RACMO2.4p1 (R24, Van Dalum et al., 2024b) has been released and includes, among other changes, an update of IFS physics parameterizations to cycle 47r1 (ECMWF, 2020). Furthermore, several processes are updated since the previous operational polar version of RACMO, R23p2, that are part of the evaluated but not operational version RACMO2.3p3 (R23p3, Van Dalum et al., 2020, 2021, 2022). Here we provide a short overview of recent updates and model settings.

### 2.1.1 Updates from RACMO2.3p3

R23p3 includes a new spectral snow albedo and radiative transfer scheme that allows for radiation penetration and subsurface heating. The Two-streAm Radiative TransfEr in Snow model (TARTES, Libois et al., 2013), which calculates radiative transfer and albedo for a given snowpack and wavelength, is included in RACMO by coupling it to the 14 available spectral bands using the Spectral-to-NarrOWBand ALbedo (SNOWBAL) module version 1.2 (Van Dalum et al., 2019). TARTES uses the geometric-optics Approximate Asymptotic Radiative Transfer (AART) theory (Kokhanovsky, 2004) and delta-Eddington approximation (Joseph et al., 1976) to solve the radiative transfer equations. The TARTES-SNOWBAL combination has replaced the broadband albedo scheme of Kuipers Munneke et al. (2011) and is also used as a new bare ice albedo scheme (Van Dalum et al., 2020). In addition, several changes are made to the multi-layer snow scheme, including a new snow layer merging routine and a higher number of layers near the surface.

### 2.1.2 Updates from RACMO2.4p1

In R24, the IFS physics parameterizations are updated from cycle 33r1 (ECMWF, 2009) to cycle 47r1 (ECMWF, 2020). Major updates have been implemented by the ECMWF between these cycles. In the cloud scheme, liquid and ice water in clouds are now treated separately as prognostic variables. Several processes are therefore represented more realistically, such as mixed-phase clouds. Rain and snow are now separate prognostic variables, allowing them to have a fall speed and to be moved by advection of the ambient air flow instead of precipitating instantaneously to the ground. More precipitation types are therefore modeled, such as wet snow and ice pellets. Aerosols are now prescribed by the Atmospheric Monitoring Service (CAMS, Bozzo et al., 2020). The radiation scheme is replaced by ecRad (Hogan and Bozzo, 2016) that includes several new parameterizations, like longwave radiation scattering by clouds, and cloud optical properties now use the Suite Of Community RAdiative Transfer codes based on Edwards and Slingo (SOCRATES). The FLake model (Mironov et al., 2010) replaces the previous rudimentary lake model, but its impact is limited due to the small number of non-supraglacial lakes in Antarctica. In addition, a multi-layer seasonal snow scheme is introduced, now allowing up to 5 layers with a more realistic representation of snow density and temperature gradients. This results in reduced thermal inertia and a more accurate diurnal cycle (Arduini et al., 2019). The IFS snow scheme is used in R24 only for seasonal snow over non-glaciated surfaces.

Other changes include updated climatological fields like background albedo and orography-related fields. For Antarctica, the ice mask is now based on BedMachine Antarctica version 3 (Morlighem et al., 2020), which is regridded to the RACMO grid. Fractional ice cover is now allowed, resulting in grid cells that can be partially glaciated. This is in particular relevant around the McMurdo Dry Valleys (Fig. 1). Furthermore, the blowing snow parameterization is revised (Gadde and Van de Berg, 2024), resulting in increased total sublimation, and the parameterization for superimposed ice is modified. A more detailed description of all implemented changes in R24 are described by Van Dalum et al. (2024b).

### 2.1.3 Model settings

In this study, R24 is run on an 11 km horizontal resolution grid for a domain that includes Antarctica, the Drake Passage and the southern tip of South America, in compliance with the Antarctic CORDEX domain (https://climate-cryosphere.org/antarctic-cordex/, last access: 19 May 2025) and daily RACMO output are used. The number and quality of observations increased considerably in the Antarctic since the International Geophysical Year of 1957-1958 (Martin, 1958), hence we run R24 for the period 1960-2023, with 1955-1960 as spinup. In comparison, R23p2 was run on a 27 km horizontal resolution grid that includes only Antarctica, and covers 1979-2023, with 1974-1979 as spinup. In the atmosphere, 40 layers are utilized and both RACMO versions are forced by ERA5 (Hersbach et al., 2020). RACMO is nudged at the upper boundary (Van de Berg and Medley, 2016) and no impurities are prescribed in the snow. The firn-column state of R24 is initialized by rasterizing a previous RACMO2.3p3 run such that as many grid points as possible have a fully-developed firn column. Other grid points that were not glaciated in previous RACMO versions are initialized with a reference snowpack, allowing a realistic firn column to form during initalization. The R23p2 run was initialized with the output of the IMAU firn densification model (IMAU-FDM, Ligtenberg et al., 2018).

### 2.2 Surface energy balance and surface mass balance

Melt can occur at the surface and inside the snowpack. Surface melt $M$ is calculated as the residual of the surface energy balance (SEB) components when the skin temperature is 0 °C:

$$M = \mathrm{LW_{net}} + \mathrm{SW_{net}} + \mathrm{SHF} + \mathrm{LHF} + G_\mathrm{s}, \tag{1}$$

with $\mathrm{LW_{net}}$ and $\mathrm{SW_{net}}$ the net longwave and shortwave radiation absorption at the surface that are defined as the sum of incoming and outgoing radiation ($\mathrm{LW_{net}} = \mathrm{LW_d} + \mathrm{LW_u}$ and $\mathrm{SW_{net}} = \mathrm{SW_d} + \mathrm{SW_u}$). Fluxes towards the surface are defined positive. SHF and LHF are the sensible and latent heat flux, respectively, and $G_\mathrm{s}$ the subsurface heat flux. Since R23p3, part of incoming shortwave radiation is absorbed below the surface, resulting in internal heating, and does not directly contribute to the SEB. Subsurface melt occurs when radiation penetration heats the snow above the melting point.

We define the SMB as the difference between accumulation and ablation in the near-surface firn or ice column, which is formally known as the climatic mass balance (Cogley et al., 2011). Accumulation is all precipitation (PR) that is retained at or near the surface, and riming (RI). Rain can add to the SMB if it freezes in the snowpack, but is not included as accumulation when it runs off, e.g., over bare ice. Ablation consists of sublimation (SU) and runoff (RU). Blowing snow erosion (ER) can be accumulative (deposition) or ablative (erosion), depending on whether the snow transport is converging or diverging.

$$\mathrm{SMB} = \mathrm{PR} + \mathrm{RI} - \mathrm{RU} - \mathrm{ER} - \mathrm{SU} \tag{2}$$

RU is modeled when layers are filled with melt water or rain and irreducible water content is reached for the snow column. In R24, SU also includes sublimating blowing snow suspended in the atmosphere (Gadde and Van de Berg, 2024).

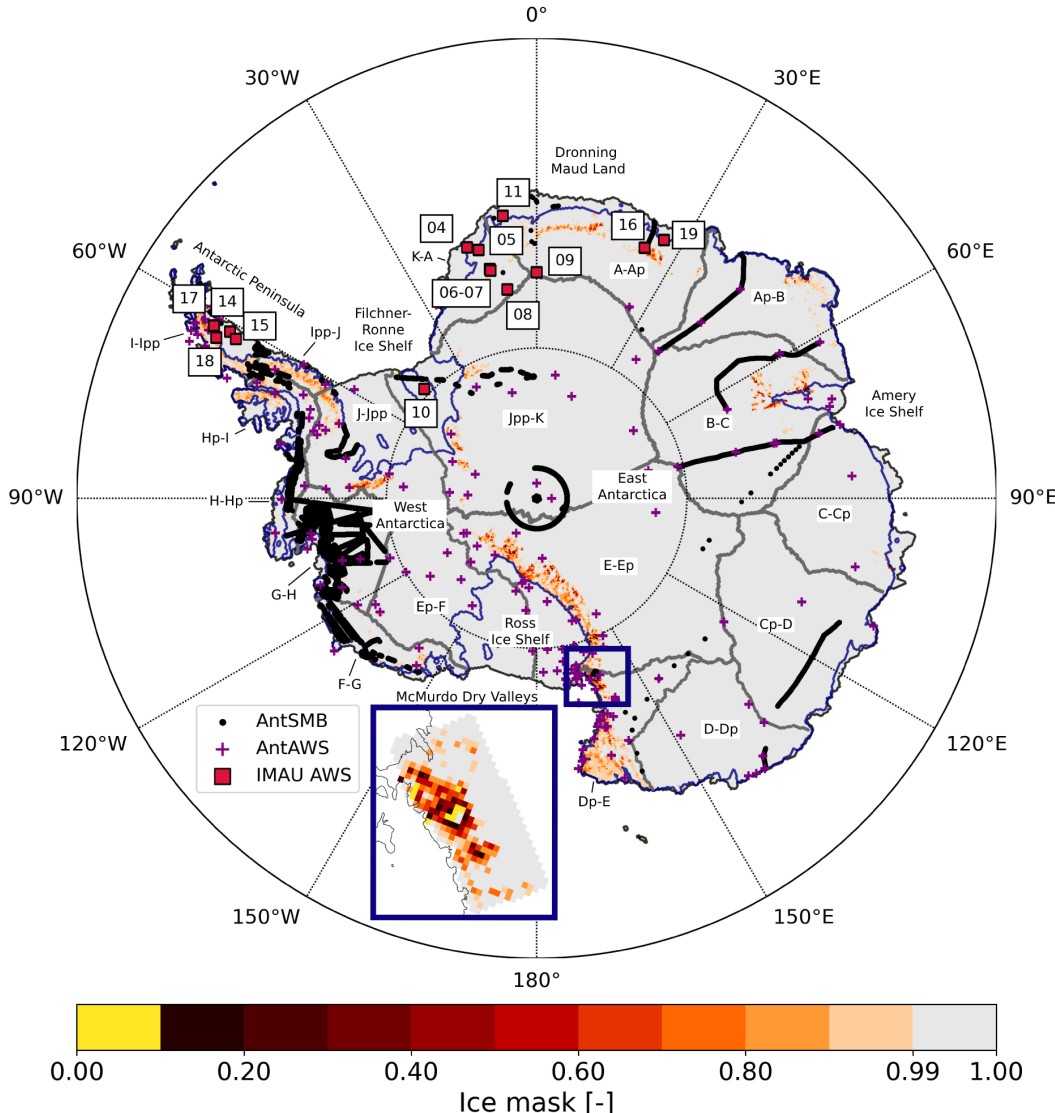

**Figure 1.** Land-ice mask of Antarctica in R24 on an 11 km horizontal resolution grid. A minimum ice fraction of 0.1 is applied, with yellow indicating ice-free grid cells. The margins of grounded ice are shown in dark blue. Measurement locations of AntSMB (Wang et al., 2021) considered in this study are shown as black dots and locations of AntAWS (Wang et al., 2023) as purple crosses. The IMAU AWSs used in this study are numbered and shown as red squares. Grid points considered in the analysis of the McMurdo Dry Valleys are highlighted in the blue box. Drainage basins are named and indicated by gray contours, following the Ice Sheet Mass Balance Inter-comparison Exercise (IMBIE) convention based on Rignot et al. (2019). The basins I-Ipp, Ipp-J and Hp-I form the Antarctic Peninsula region; H-Hp, J-Jpp, G-H, F-G and Ep-F form the West Antarctica region and the remaining basins form the East Antarctica region. Dronning Maud Land and The Filchner-Ronne, Ross and Amery ice shelves are also indicated.

### 2.3 In situ and remote sensing data sets

#### 2.3.1 AntSMB

The AntSMB data set is a comprehensive compilation of in situ SMB observations from the literature and public data platforms (Wang et al., 2021), covering several centuries. This data set includes stake measurements, ice cores, snow pits, ground pene-trating radar and ultrasonic sounders. Many coastal zones of the AIS, including ice shelves, are covered well, both spatially and temporally, with highest coverage in West Antarctica. Large parts of the interior of the AIS, however, remain undocumented, in particular the interior plateau of East Antarctica. Several criteria are applied to the data, including a minimum duration of records, checks whether all essential data parameters like location and measurement methodology are provided, and quality-control checks. Locations of in situ SMB measurements used in this study are shown in Fig. 1 and will also be categorized by drainage basin following the Ice Sheet Mass Balance Inter-comparison Exercise (IMBIE) convention, which is based on Rignot et al. (2019).

#### 2.3.2 AntAWS

A compilation of automatic weather station (AWS) data is published recently that includes all available Antarctic AWS (AntAWS) records from 1980 to 2021 (Wang et al., 2023). This data set includes air temperature, humidity, air pressure, wind direction and wind speed. Most AWSs that are part of this data set are similar in design by using a standard set of equipment. The observation height above the surface varies between 1 and 6 m, and can vary during the season. Several quality-control checks are applied, including filtering out of instrument failures due to e.g. sensor freezing or frost cover. The AntAWS stations used in this study are shown in Fig. 1. Some AntAWS observations are assimilated into ERA5, but a comparison with RACMO is appropriate, as RACMO calculates its own climate.

#### 2.3.3 IMAU AWS for SEB components

The IMAU AWSs on the AIS are set up with equipment to measure radiative fluxes, and turbulent fluxes are estimated using the measured temperature, humidity and a constant aerodynamic roughness of 0.1 mm for snow and 1 mm for ice (Van Tiggelen et al., 2023). Similarity theory flux-gradient relations with stability correction functions described by Holtslag and Bruin (1988) are used to correct measurements to standard heights. The processing steps of the IMAU AWS data are further described by Smeets et al. (2018) and Van Tiggelen et al. (2024). In total, 151 station-years of hourly data are available between 1995 and 2022 from 19 stations. The stations used in this study are shown in Fig. 1. Observations are not assimilated into ERA5 and are daily averaged for downward and upward shortwave and longwave radiative fluxes, turbulent fluxes, temperature, wind speed and surface pressure.

Most stations are located on the AP and in Dronning Maud Land (DML), except for AWS 10, which is located on Berkner Island near the Filchner-Ronne ice shelf (Fig. 1). In the AP, AWS 14, 15 and 18 are located on the Larsen C ice shelf. AWS 18 is located close to the grounding line and the AP mountain range, and föhn wind events occur frequently. AWS 17 is located on

the remnants of the Larsen B ice shelf in Scar Inlet. AWS 4 and 19 are located on ice shelves in DML close to the grounding line, while AWS 5 to 9 and 11 are located on grounded ice in western DML. In eastern DML, AWS 16 is located close to exposed mountains where the ice mask of RACMO is fractional (Fig. 1).

### 2.3.4 MODIS albedo product

The Moderate Resolution Imaging Spectroradiometer (MODIS), version 006 of MCD34A3 (Schaaf and Wang, 2015) provides a daily broadband white-sky albedo product at local solar noon based on observations of seven shortwave bands with a resolution of 500 m (250 m for band 1 and 2). The accuracy of MODIS albedo, however, drops with solar zenith angles (SZAs) of 55° or higher, and becomes unrealistic for SZAs of 65° (Liu et al., 2009; Wang and Zender, 2010). Furthermore, Manninen et al. (2019) and Wang and Zender (2010) noted that the albedo for too high latitudes (>75° S) should be excluded, as the signal-to-noise ratio becomes too high, reducing its reliability in Antarctica. In this study, we therefore only use MODIS for case studies on ice shelves surrounding the East Antarctic ice sheet (EAIS) during the summer of 2004-2005.

Under overcast conditions, radiation loses its solar zenith angle dependence, as the direct radiation component is lost and the snow surface is illuminated diffusely. In addition to a spectral shift towards visible light that typically occurs for cloudy conditions, for which the spectral albedo is high (Dang et al., 2015), the resulting broadband albedo increases with cloud cover and is higher than those at any solar zenith angles for clear-sky conditions (Liljequist, 1956; Aoki et al., 1999). These effects are not included in the MODIS white-sky albedo product, and some discrepancies are therefore expected when comparing with the only albedo available in RACMO, which is calculated by dividing the outgoing with the incoming solar radiation at the surface and includes both direct and diffuse radiation.

### 2.3.5 Passive microwave radiometer observations of liquid water in snow

Spaceborne passive microwave radiometers measure the brightness temperature of a snowpack. The presence of liquid water in snow causes a sharp change in the microwave signal, hence allowing the detection of liquid water, resulting in a wet/dry remote sensing product. The Special Sensor Microwave/Imager (SSM/I), Advanced Microwave Scanning Radiometer - Earth Observing System sensor (AMSR-E) and Advanced Microwave Scanning Radiometer 2 (AMSR2) on board satellites observe microwaves passively at several frequencies. Picard (2022) converted these observations at 19 GHz frequency with horizontal polarization to a daily wet/dry signal, based on an algorithm developed by Picard and Fily (2006) and Torinesi et al. (2003). Data are available between 1979-2021.

Several uncertainties arise with the usage of passive microwave radiometers for observing liquid water in snow (Picard, 2022). The minimum amount of liquid water observable by radiometers is not well known (Tedesco et al., 2007) and varies with snow conditions (grain size, snow density, etc.). Here we apply a threshold of 0.5 kg m$^{-2}$ following recommendations of Picard et al. (2022). Up to what depth liquid water can be measured is also depending on snow conditions, and changes during the melt season due to snow metamorphism. More importantly, however, are uncertainties due to the observation hour, as AMSR-E and AMSR2 measure around local noon and midnight, while SSM/I measures during the early morning and evening. As melt water often refreezes overnight, some melt events can therefore be potentially missed. This is corrected for in the data

as is discussed in more detail by Picard and Fily (2006). Other uncertainties, such as varying resolution and angle of incidence between satellites, are deemed to be small (Picard et al., 2022; Banwell et al., 2023).

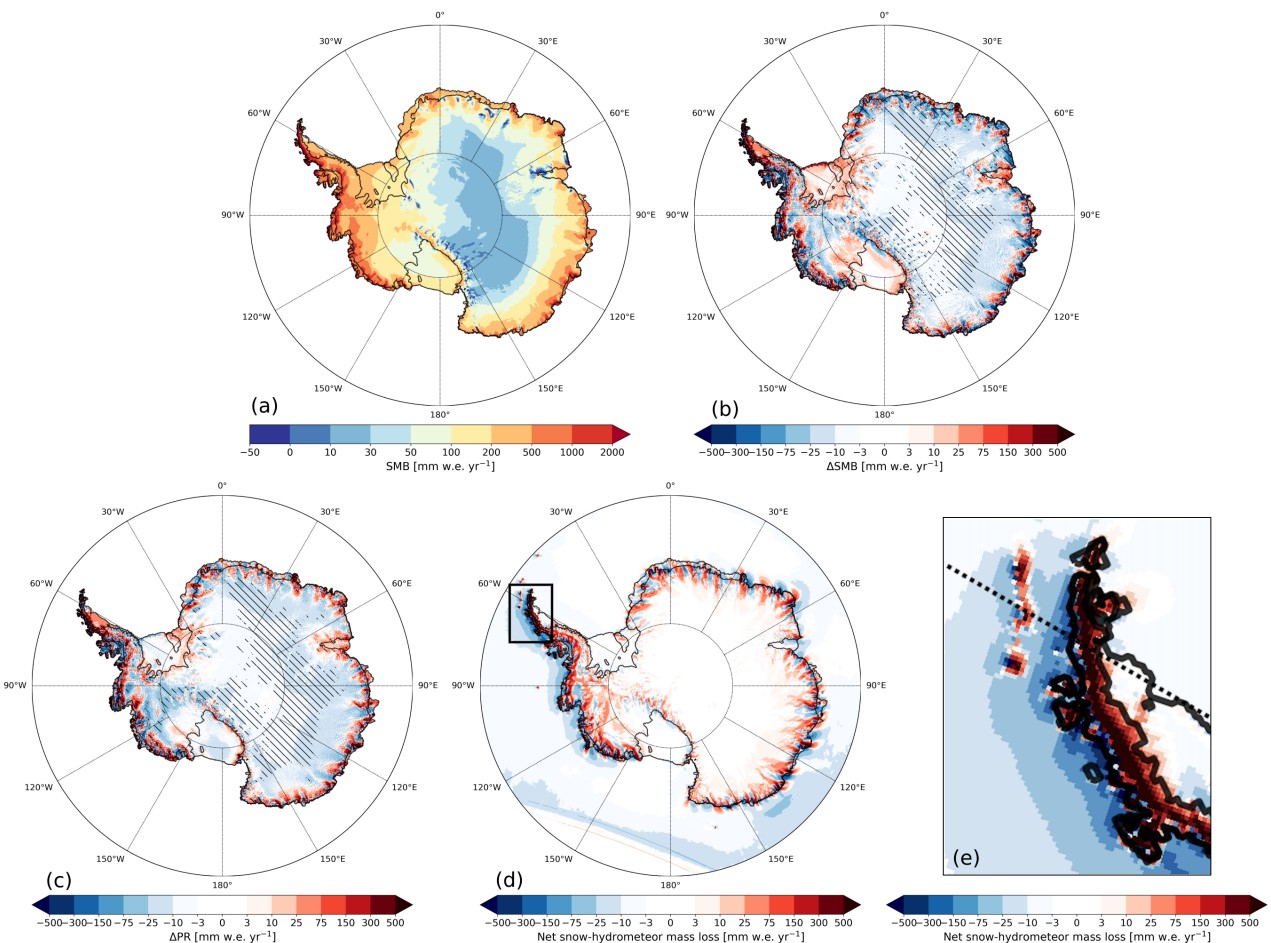

**Figure 2.** Annual accumulated SMB of R24 **(a)** for 1979-2023 in mm water equivalent (w.e.) yr$^{-1}$. The difference with respect to R23p2 for **(b)** SMB and **(c)** precipitation. Hatched areas indicate differences that exceed inter-annual variability. Snow-hydrometeor mass loss as is determined in the IFS cloud scheme in R24 for **(d)** the AIS and **(e)** the northern part of the AP. Red indicates mass loss, blue mass gain.

### 2.3.6 QuikSCAT melt product

The SeaWinds scatterometer aboard the QuikSCAT (QSCAT) satellite provides a radar backscatter signal that can also be used to detect liquid water in snow. Trusel et al. (2013) developed a method to produce a seasonal total melt water flux product 220 by calibrating the QSCAT data with in situ observations from AWSs that enabled SEB closure and melt rate quantification. Daily data are available between 2000 and 2009 around austral summer and are converted to the RACMO grid, with e.g. 2003 referring to the 2002-2003 ablation season.

## 3 Results: Surface mass balance

### 3.1 AntSMB comparison

Figure 2a shows the annual accumulated SMB of R24. Virtually all of the AIS ice sheet is characterized by a positive SMB, with particularly high values on the coast of West Antarctica and the western side of the AP (Fig. 3b and Fig. 3c, up to 2000 mm w.e. yr$^{-1}$). The eastern side of the AP, which is separated climatologically from the western side by a narrow mountain range, is characterized by a considerably lower SMB, in particular on the Larsen C ice shelf (often lower than 500 mm w.e. yr$^{-1}$). The coast of East Antarctica also has a high SMB in general, while the interior is dry and the SMB is much lower (Fig.

3d, often lower than 30 mm w.e. yr$^{-1}$). The only places with a (near-)zero SMB are in the western part of the Amery ice shelf near the grounding line, where exposed blue ice is modeled that lowers the albedo and where melt water ponds are frequently observed (Tuckett et al., 2021). Small areas in eastern DML and inland of the Transantarctic mountains in East Antarctica are not glaciated and have exposed bedrock.

With respect to the majority of in situ observations used in this study between 1979-2017 of the AntSMB data set, which
cover the entire AIS but mostly originate from radar observations performed on the WAIS (145,427 observations), the SMB of R24 is modeled well (Fig. 3a) and the linear regression fit (red line) is close to the 1-on-1 line (black line). The fit, however, deviates more for high SMB values (>1500 mm water equivalent (w.e.) yr$^{-1}$), where R24 tends to underestimate the SMB. This happens more frequently in the AP (Fig. 3c) and the WAIS (Fig. 3b), where not enough precipitation is modeled for high-accumulation areas. For the EAIS, differences are smaller (Fig. 3d), as observations derive mostly from dry locations.
The SMB of R24 shows improvements compared R23p2 (Fig. A1, Table 1), as the root-mean-square error (RMSE; 201.4 and 227.2 mm w.e. yr$^{-1}$ for R24 and R23p2, respectively), determination coefficient (R$^2$; 0.73 and 0.66) and bias (-28.3 and -55.3 mm w.e. yr$^{-1}$) have improved. In particular the underestimation of the SMB in R23p2 in high-accumulation areas in the WAIS and the AP is reduced considerably. In the EAIS, the RMSE and R$^2$ have improved, while the SMB is now somewhat too low. For the ice shelves, the RMSE, bias and R$^2$ have improved considerably.

Several observational sites are in close proximity to each other with measurements done in short succession, for which the SMB of the same RACMO grid point is used, resulting in horizontal 'stripes' in Fig. 3. Each 'stripe' therefore illustrates subgrid variability. Improving the horizontal resolution further would diminish this effect. Increasing the resolution in R24, however, is not the primary cause of the improvements between R24 and R23p2. When eliminating subgrid variability by first calculating the mean of all observations available for a grid point, followed by using this mean for all observations of the considered grid
point, results in an RMSE of 145.4 and 166.5 mm w.e. yr$^{-1}$ for R24 and R23p2, respectively, while the bias remains the same. As the RMSE difference between R24 and R23p2 remains roughly equal, eliminating subgrid variability therefore shows that improvements of the SMB in R24 with respect to R23p2 are not primarily caused by increased horizontal resolution.

### 3.2 Changes compared to R23p2

Compared to R23p2 (Fig. 2b), the SMB has changed around the margins of the AIS and in the interior, where changes ex-
ceed inter-annual variability. Most SMB differences are caused by precipitation changes (Fig. 2c). In the interior of the EAIS,

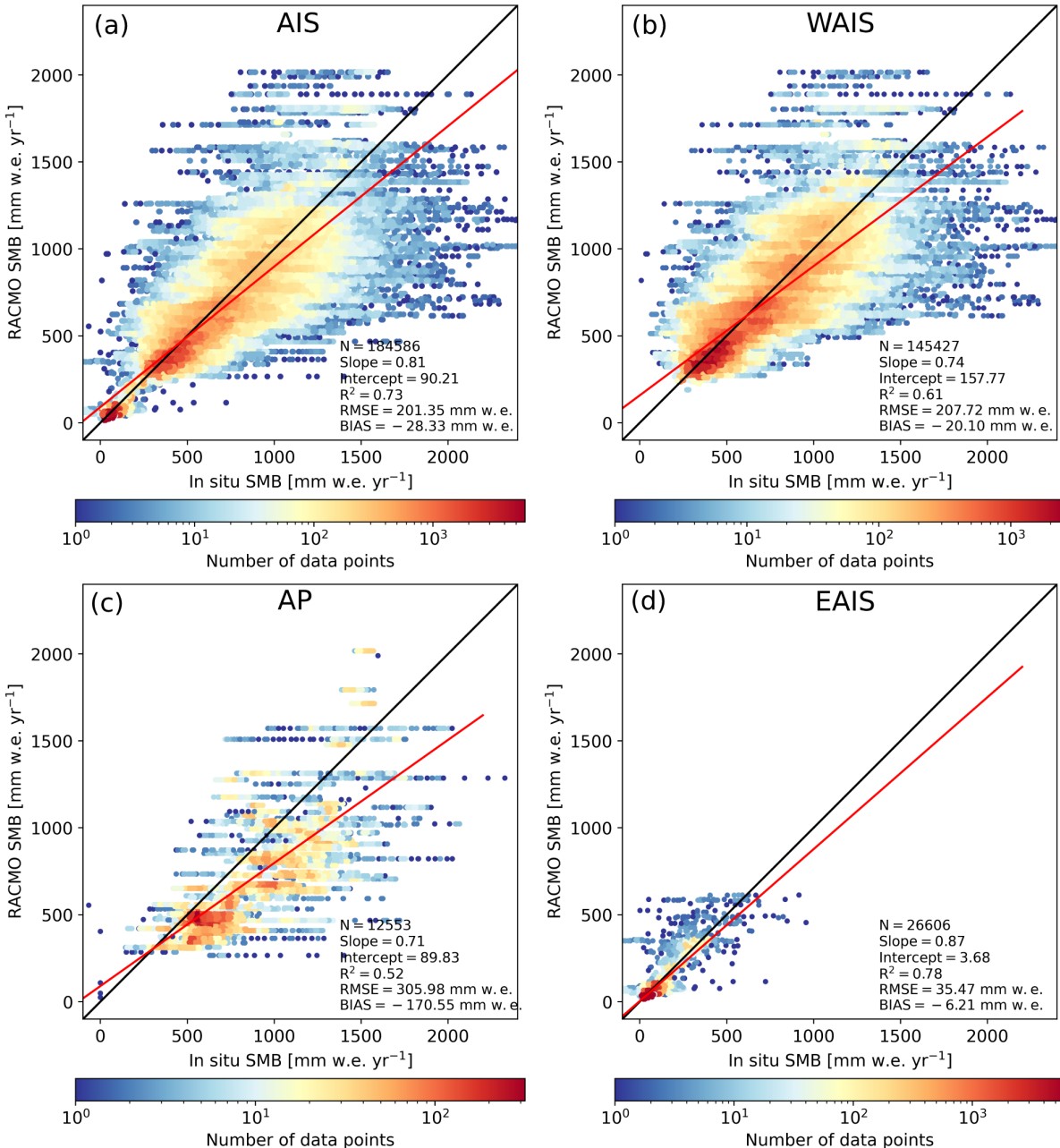

**Figure 3.** SMB of R24 with respect to observations of the AntSMB data set for 1979-2017 for **(a)** the entire AIS, **(b)** the WAIS, **(c)** the AP and **(d)** the EAIS. The 1-on-1 line is shown in black and the linear regression fit in red. The number of observations (N), slope and intercept of the linear regression fit, determination coefficient ($R^2$), root-mean-square error (RMSE) and bias are also shown. AntSMB measurement locations are shown in Fig. 1.

**Table 1.** Integrated SMB for the AIS, including ice shelves, for R24, R23p2 and the difference between R24 and R23p2 ($\Delta$) in Gt yr$^{-1}$ and in percentage (1979-2023). Ice shelves considered separately and drainage basins, including ice shelves and grounded ice following the IMBIE convention based on Rignot et al. (2019) (illustrated in Fig. 1), are shown as well. The drainage basins are categorized by the EAIS, WAIS and the AP. Total number of observations (N), determination coefficient ($R^2$), root-mean-square error (RMSE) and bias for R24 and R23p2 with respect to (w.r.t.) AntSMB observations (1979-2017) are also shown for each category. The RMSE and bias are in mm w.e. yr$^{-1}$. AntSMB measurement locations are shown in Fig. 1.

| Basin | Integrated | | | | R24 w.r.t. observations | | | | R23p2 w.r.t. observations | | | |
|---|---|---|---|---|---|---|---|---|---|---|---|---|
| | R24 | R23p2 | $\Delta$R23p2 | $\Delta$% | N | $R^2$ | RMSE | Bias | N | $R^2$ | RMSE | Bias |
| **AIS** | 2545.7 | 2604.4 | -60.7 | -2.3 | 184586 | 0.73 | 201.4 | -28.3 | 184608 | 0.66 | 227.2 | -55.3 |
| **Ice Shelves** | 578.9 | 551.8 | 27.1 | 4.9 | 15316 | 0.69 | 182.3 | -19.3 | 14928 | 0.66 | 196.3 | -51.4 |
| **East** | 1312.5 | 1413.5 | -101.0 | -7.1 | 26606 | 0.78 | 35.5 | -6.2 | 26606 | 0.73 | 37.8 | 2.6 |
| Jpp-K | 151.7 | 154.5 | -2.8 | -1.8 | 16197 | 0.92 | 19.8 | -6.9 | 14610 | 0.88 | 24.7 | -3.7 |
| K-A | 75.2 | 77.3 | -2.1 | -2.7 | 28 | 0.01 | 372.7 | 361.1 | 28 | 0.14 | 333.9 | 317.9 |
| A-Ap | 137.9 | 165.9 | -28.0 | -16.9 | 74 | 0.43 | 152.8 | -39.2 | 74 | 0.25 | 171.3 | -87.6 |
| Ap-B | 123.4 | 134.2 | -10.8 | -8.0 | 584 | 0.56 | 46.0 | 3.59 | 554 | 0.55 | 63.7 | 29.3 |
| B-C | 89.8 | 99.9 | -10.1 | -10.1 | 958 | 0.32 | 39.7 | -9.4 | 963 | 0.34 | 38.6 | 1.61 |
| C-Cp | 170.8 | 181.7 | -10.9 | -6.0 | 88 | 0.30 | 68.3 | 30.6 | 104 | 0.51 | 56.7 | 0.16 |
| Cp-D | 246.0 | 255.5 | -9.5 | -3.7 | 367 | 0.28 | 87.7 | 21.7 | 367 | 0.21 | 92.8 | 27.3 |
| D-Dp | 156.7 | 160.9 | -4.2 | -2.6 | 93 | 0.07 | 219.8 | 108.0 | 91 | 0.04 | 257.9 | 150.5 |
| Dp-E | 47.5 | 50.3 | -2.8 | -5.6 | 176 | 0.13 | 204.3 | 187.4 | 66 | 0.17 | 188.2 | 162.5 |
| E-Ep | 113.5 | 133.3 | -19.8 | -14.9 | 8041 | 0.62 | 21.4 | -13.5 | 9749 | 0.37 | 29.2 | 7.04 |
| **West** | 832.6 | 818.1 | 14.5 | 1.8 | 145427 | 0.61 | 207.7 | -20.1 | 145136 | 0.52 | 233.5 | -50.3 |
| Ep-F | 105.5 | 101.9 | 3.6 | 3.5 | 9 | 0.00 | 86.6 | -86.5 | 0 | - | - | - |
| F-G | 194.0 | 189.1 | 4.9 | 2.6 | 15578 | 0.12 | 313.6 | 25.4 | 15305 | 0.13 | 313.4 | 65.7 |
| G-H | 217.4 | 223.5 | -6.1 | -2.7 | 110771 | 0.59 | 169.7 | -22.5 | 108935 | 0.50 | 191.2 | -47.6 |
| H-Hp | 129.9 | 131.8 | -1.9 | -1.4 | 14429 | 0.22 | 324.7 | -51.2 | 16256 | 0.35 | 383.1 | -179.8 |
| J-Jpp | 185.8 | 171.8 | 14.0 | 8.1 | 4640 | 0.48 | 82.8 | -18.6 | 4640 | 0.34 | 99.1 | -42.8 |
| **Peninsula** | 400.7 | 374.7 | 26.0 | 6.9 | 12553 | 0.52 | 306.0 | -170.6 | 12866 | 0.48 | 349.6 | -231.8 |
| Hp-I | 177.6 | 180.6 | -3.0 | -1.7 | 6073 | 0.19 | 339.4 | -183.9 | 5513 | 0.17 | 336.2 | -150.3 |
| I-Ipp | 193.6 | 159.4 | 34.2 | 21.5 | 3289 | 0.60 | 332.3 | -201.9 | 2825 | 0.68 | 414.9 | -367.5 |
| Ipp-J | 29.5 | 34.7 | -5.2 | -15.0 | 3191 | 0.22 | 185.9 | -112.8 | 4528 | 0.0 | 319.8 | -246.3 |

precipitation is lower in R24, which is likely due to revisions in the cloud scheme. Around the coast of East Antarctica, an alternating pattern of increased and decreased SMB is apparent, with lower SMB on the lower-lying ice streams and higher SMB on the higher-elevated ridges. The advection of snow hydrometeors causes this pattern, as it enhances orographic precipitation while reducing precipitation on lower-lying ice streams. Figure 2d illustrates this effect by showing the average net

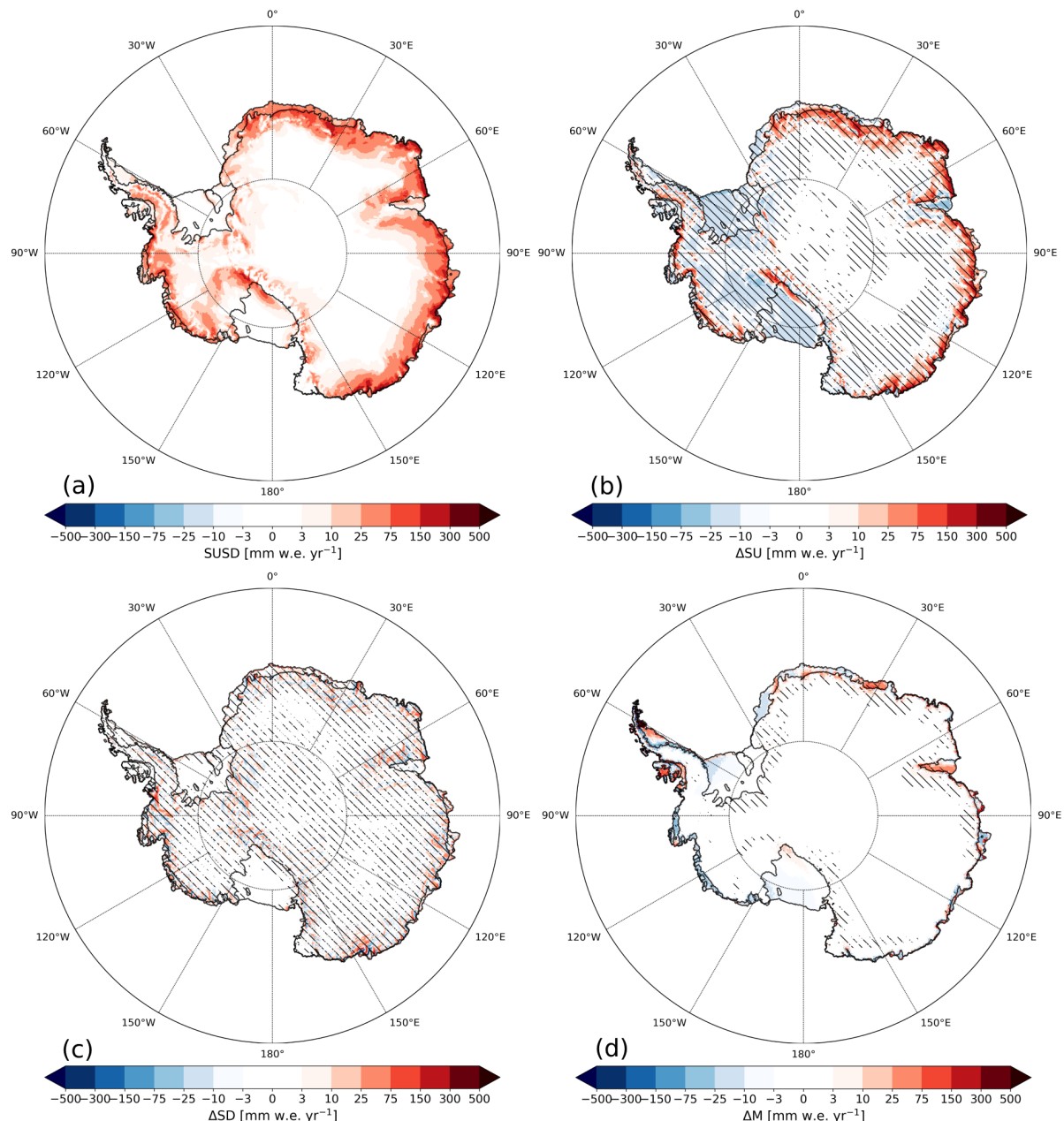

**Figure 4. (a)** Annual accumulated sublimation of suspended snow in the atmosphere due to blowing snow in R24 for 1979-2023. The difference with respect to R23p2 for **(b)** total sublimation, **(c)** blowing snow erosion and **(d)** melt. Hatched areas indicate differences that exceed inter-annual variability.

mass loss or gain of snow hydrometeors, with positive values indicating more mass loss than local production. Precipitation can only exceed local snow hydrometeor production if snow hydrometeors have moved in from elsewhere, and Fig. 2d there-

**Table 2.** Integrated surface mass balance (SMB), precipitation (PR), melt (ME), refreezing (RF), runoff (RU), blowing snow erosion (ER) and sublimation (SU) over the AIS, including ice shelves, for R24, R23p2, the difference between R24 and R23p2 ($\Delta$) and the standard deviation ($\sigma$) for 1979-2023 in Gt yr$^{-1}$. In R24, SU includes sublimation of blowing snow suspended in the atmosphere. Differences in percentages are also shown.

| Variable | R24 | R23p2 | $\Delta$R23p2 | $\Delta$% | $\sigma_{\text{R24}}$ | $\sigma_{\text{R23p2}}$ |
|---|---|---|---|---|---|---|
| SMB | 2545.7 | 2604.4 | -60.7 | -2.3 | 133.3 | 125.5 |
| PR | 2798.1 | 2790.7 | 7.4 | 0.3 | 134.9 | 125.5 |
| ME | 123.7 | 117.8 | 5.9 | 5.0 | 31.2 | 30.5 |
| RF | 122.5 | 119.3 | 3.2 | 2.7 | 31.1 | 29.5 |
| RU | 5.0 | 4.1 | 0.9 | 22.0 | 2.5 | 2.1 |
| ER | 11.2 | 5.4 | 5.8 | 107.4 | 0.6 | 0.2 |
| SU | 236.2 | 175.4 | 60.8 | 34.7 | 9.8 | 8.0 |

fore effectively illustrates the net effect of horizontal transport of snow hydrometeors on modeled precipitation. R23p2 has no horizontal advection of hydrometeors and all precipitation instantaneously reaches the surface. Comparing Fig. 2c with Fig.2d shows that most of the precipitation changes are due to the advection of snow hydrometeors. This advection also leads to

increased precipitation and SMB in mountaineous areas, in particular in West Antarctica and the AP. Snow hydrometeors that are formed above the ocean are now moved onshore where they orographically precipitate in the mountains while lowering precipitation off the coast on the ocean (Fig. 2e). Because of the much higher fall speeds and low absolute amounts, the effect of the advection of rain is generally negligible on the AIS. The introduction of advection of hydrometeors brings the modeled precipitation patterns qualitatively closer to that of the MAR regional climate model (Fig. 4 of Fettweis et al. (2020)), where

advection of hydrometeors is already calculated since several model versions.

At the Ross and Filchner-Ronne ice shelves, decreased sublimation (Fig. 4b) has led to increased SMB with respect to R23p2 (Fig. 2b), despite lower precipitation (Fig. 2c). West Antarctica, and the Ross and Filchner-Ronne ice shelves in particular, are characterized by lower temperatures (discussed in Sect. 5), consequently reducing sublimation. Sublimation has not changed considerably in the interior of the EAIS. Around the margins of the AIS, however, sublimation has increased significantly

despite lower temperatures. In R23p2, mass and energy from sublimated suspended snow was extracted from the surface, while in R24 suspended snow sublimation directly moistens the atmospheric layer in which the sublimation occurs (Fig. 4a). As snow drift sublimation occurs primarily at the top of the snow drift column, it occurs tens to, in extreme cases, hundreds of meters above the surface. The updated snow drift physics agrees better with observations and results in more sublimation (Gadde and Van de Berg, 2024). This is in particular relevant in regions with strong katabatic winds that typically occur around

the margins of the AIS (Parish and Cassano, 2003). Blowing snow transport, however, is still small despite significant changes compared to R23p2 for most regions. Melt in R24 is often small and comparable to R23p2, but spatial differences are significant for several regions. This is discussed in more detail in Sect. 4.

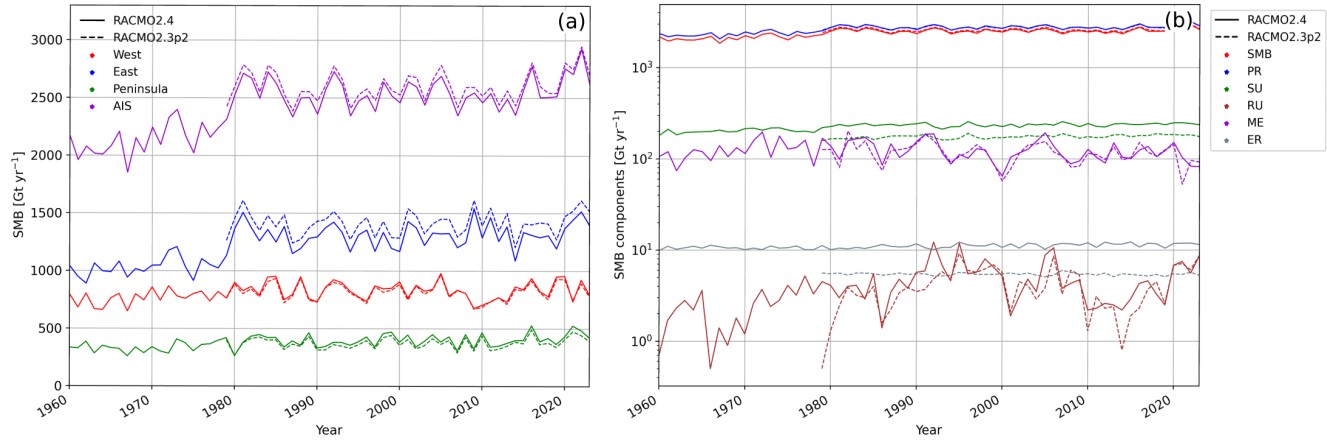

**Figure 5.** Time series of **(a)** yearly domain- and regional-integrated SMB for R24 (1960-2023) and R23p2 (1979-2023), including ice shelves, and **(b)** SMB components for the entire AIS, including precipitation (PR), sublimation (SU), runoff (RU), melt (ME) and blowing snow erosion (ER) on a logarithmic scale.

## 3.3 Integrated surface mass balance

Integrated over the ice sheet, the SMB in R24 has only marginally decreased with respect to R23p2 (-60.7 Gt yr$^{-1}$, -2.3%,
Table 2), which is below the standard deviation. Precipitation has increased slightly (7.4 Gt yr$^{-1}$, 0.3%). The aforementioned alternating patterns of increased and decreased precipitation shown in Fig. 2c thus average out when integrated over the ice sheet. Melt and runoff changes are small (5.9 and 0.9 Gt yr$^{-1}$, respectively), but by percentage runoff has increased considerably (22.0%). Blowing snow erosion has increased significantly due to higher wind speeds over the AIS (107.4%), but it still relatively small (11.2 Gt yr$^{-1}$), as blowing snow erosion only reduces the integrated SMB when snow is blown off the ice sheet. Most of the SMB change is caused by a significant increase in sublimation (60.8 Gt yr$^{-1}$, 34.7%). Of the 236.2 Gt yr$^{-1}$ sublimation modeled by R24, 223.3 Gt yr$^{-1}$ is due to sublimation of blowing snow. Gadde and Van de Berg (2024) report on average a sublimation of blowing snow of 175 Gt yr$^{-1}$ in their development version of RACMO2.3p3, which is higher than the calculated 101 Gt yr$^{-1}$ in the regional climate model CRYOWRF. Gadde and Van de Berg (2024) employed RACMO2.3p3 on a horizontal resolution grid of 27 km and reported that compared to satellite observations of Palm et al. (2017), underestimation of blowing snow is possible near the coast as the resolution is not high enough to capture the large spatial gradients. The horizontal resolution of R24, however, is 11 km. In addition, Gadde and Van de Berg (2024) report a blowing snow erosion of 8 Gt yr$^{-1}$, which is lower than 11.2 Gt yr$^{-1}$ in R24, and the larger integrated sublimation of blowing snow is therefore expected.

Table 1 shows that the SMB, including ice shelves, has increased by 6.9% (26.0 Gt yr$^{-1}$) for the AP and 1.8% (14.5 Gt yr$^{-1}$) for the WAIS, while it has decreased for the EAIS by 7.1% (-101.0 Gt yr$^{-1}$). Most of the SMB signal is therefore originating from the EAIS. For the ice shelves when considered separately, the SMB has increased by 4.9% (27.1 Gt yr$^{-1}$).

A time series of the SMB for the EAIS, WAIS, AP and the entire AIS illustrates that inter-annual variability is similar in both model versions (Fig. 5a), which is expected as they are both forced by ERA5 at the lateral boundaries. Compared to R23p2, the SMB of the entire AIS is lower for each year in R24, despite higher SMB in the AP. Before 1979, the SMB of R24 is considerably lower, as moisture is less constrained in reanalysis due to the lack of satellite observations, resulting in reduced precipitation on the EAIS (Tietäväinen and Vihma, 2008; Van de Berg et al., 2005). The precipitation jump coincides with the onset of Television Infrared Observation Satellite (TIROS) Operational Vertical Sounder (TOVS) measurements in late 1978, mounted on TIROS satellites (Bromwich and Fogt, 2004). We therefore deem pre-1979 SMB unreliable. The other SMB components do not show significant trends (Fig. 5b) and are at least an order of magnitude smaller than precipitation. In recent years, the SMB has increased on the AIS (Fig. 5a), in particular for 2022, which mostly originates from mass gain on the EAIS.

The integrated SMB of R24 is comparable to that of other regional climate models. Mottram et al. (2021) show that MARv3.10 and HIRHAM5 0.11° model a higher integrated SMB for the AIS including ice shelves of 2633 and 2657 Gt yr$^{-1}$, respectively, for the period 1980 to 2010. MetUM and COSMO-CLM, however, model lower SMB values, with 2327 and 2023 Gt yr$^{-1}$. Mottram et al. (2021) note that differences in horizontal resolution are the main cause of intermodel SMB differences, in particular in the AP and WAIS, where steep slopes cause orographic precipitation. Qualitatively comparing Fig. 2a to the ensemble mean of Fig. 6a of Mottram et al. (2021) shows that the SMB of R24 is also spatially in good agreement, and has improved in particular on the coast of the EAIS compared to RACMO2.3 (Fig. 6e of Mottram et al. (2021)).

### 3.4 SMB of drainage basins

Splitting the AIS in drainage basins reveals that the SMB has changed considerably for several basins (Table 1, where 1960-1979 are excluded from the statistics). In the EAIS, all domains have lowered SMB with respect to R23p2, but several domains stand out. In A-Ap, which is located in DML, and Ap-B, which is adjacent to A-Ap to the east (Fig. 1), the SMB has decreased by 16.9 and 8.0%, respectively. The observations of the AntSMB data set show that the SMB has improved in R24 for A-Ap and Ap-B, as the $R^2$, RMSE and bias have improved. The SMB of E-Ep, which includes large areas of the East Antarctic Plateau, is underestimated, resulting in a negative bias, but improvements are apparent in the RMSE (21.4 and 29.2 mm w.e. yr$^{-1}$, for R24 and R23p2 respectively) and in particular the $R^2$ (0.62 and 0.37). For B-C that also covers a considerable part of the East Antarctic Plateau, the SMB is now underestimated. The interior of the EAIS thus seems to be too dry in R24.

The SMB of the northern part of the AP (I-Ipp) is substantially higher in R24 (21.5%, 34.2 Gt yr$^{-1}$). The bias and RMSE have improved considerably, suggesting that the increased precipitation in R24 is correct for this region (Fig. 2c). For Ipp-J, which covers the south-east of the AP including Larsen D ice shelf and the AP mountain range, the SMB is now higher and closer to observations (185.9 and 319.8 mm w.e. yr$^{-1}$ RMSE; -112.8 and -246.3 mm w.e. yr$^{-1}$ bias for R24 and R23p2, respectively). For all domains in the AP, the SMB is still too low, despite increased precipitation in R24.

Most observations of AntSMB are located in the WAIS, in particular in the G-H basin that includes the Pine Island and Thwaites glaciers (110,771 valid observations for R24). The bias has become less negative in R24 (-22.5 and -47.6 mm w.e. yr$^{-1}$ for R24 and R23p2, respectively), despite a lower domain-integrated SMB (-2.7%). The $R^2$ and RMSE have also improved

considerably (0.59 and 0.50; 169.7 and 191.2 mm w.e. yr$^{-1}$ for R24 and R23p2, respectively). Local patterns are thus resolved better in R24. The J-Jpp basin provides valuable information about the grounding line of the Evans ice stream that feeds into the Ronne ice shelf, as many observations are close to it (Fig. 1). The SMB of R24 for J-Jpp has increased considerably with respect to R23p2 (8.1%) and matches well with observations.

## 4 Melt

### 4.1 Number of days with liquid water in snow


The presence of liquid water in a snowpack can be reliably measured, even small amounts, using spaceborne passive microwave radiometers (Picard et al., 2022). The data set provided by Picard (2022) includes SSM/I, AMSR-E and AMSR2 data to determine the wet/dry status of the snowpack (Sect. 2.3.5). It covers almost the entire satellite era (1979-2021), providing valuable information about the temporal and spatial occurrence of melt water.

Figure 6a shows the 1980-2021 average number of days in a year with liquid water as is observed by the remote-sensing product of Picard (2022). Most of the coastline and ice shelves surrounding the AIS have liquid water in the snow at some point during the year, including the Ross and Filchner-Ronne ice shelves. Virtually no liquid water is observed in the interior of the ice sheet, except for some relatively small areas in the WAIS. Note that there are no observations (grey in Fig. 6a) in the elevated ice sheet interior locations where no melt liquid water is expected. Most wet days occur on the ice shelves of the AP;

in particular on the Wilkins, George VI and Larsen C ice shelves, with often more than two months of liquid water present in the snowpack. Ice shelves in West Antarctica and the ice shelves in DML, the Amery, West and Shackleton ice shelves in East Antarctica also have a considerable number of wet days.

Qualitatively, the spatial pattern and amount of days with liquid water are remarkably similar to observations in R24 (Fig. 6b). At the Wilkins ice shelf and several grid cells in the AP, however, the number of wet days is overestimated by more than

a month (Fig. 6c). This is due to the presence of aquifers in R24 below the surface layers. In the Wilkins ice shelf, these aquifers exist for several months before refreezing during winter in R24, and are typically located at least 1 m below the surface. Such aquifers are therefore undetectable by the remote sensing technique used here. The number of days with liquid water are somewhat underestimated on the ice shelves of East Antarctica, but have improved compared to R23p2, in particular on the King Baudouin and Amery ice shelves (Fig. 6d). Biases have also decreased on the ice shelves of West Antarctica,

where too many wet days are modeled in R23p2. Averaged over the AIS, more wet days occur in R24 compared to R23p2 despite a smaller refreezing grain size. Radiative transfer in R24 could play a role by extending the depth where melt occurs, increasing the amount of liquid water and delaying the complete refreezing of the snowpack (Van Dalum et al., 2022). Another possible explanation is the increased resolution in R24, leading to a better representation of orography that results in more clearly-defined ice shelves and ice streams where melt can occur.

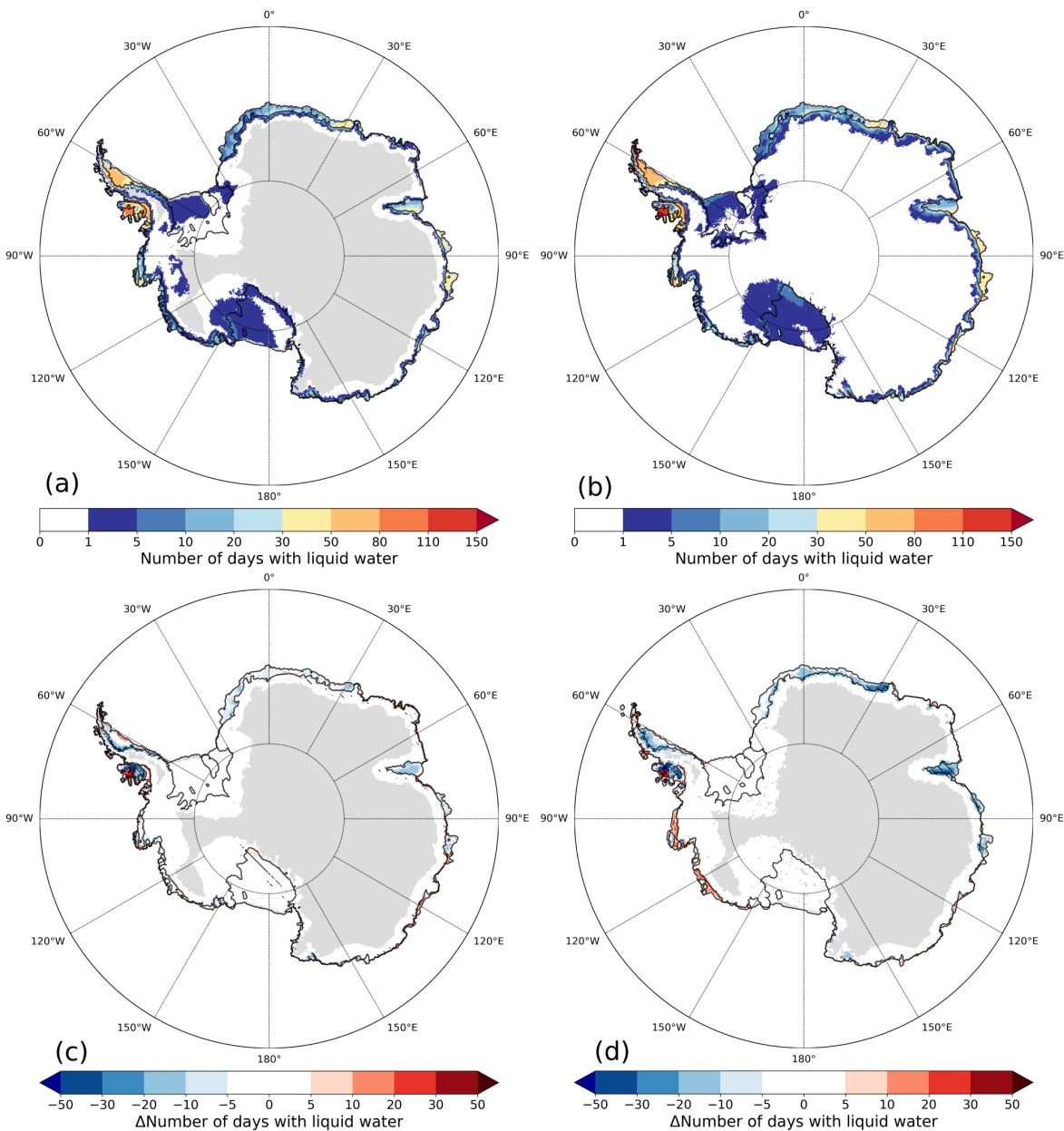

**Figure 6.** Average number of days in a year with liquid water in the snowpack (1980-2021) for **(a)** a passive microwave remote sensing product, composed of SSM/I (1980-2002), AMSR-E (2003-2011) and AMSR2 (2013-2021) observations, and **(b)** R24. Difference of number of days with liquid water with observations are shown for **(c)** R24 and **(d)** R23p2. Hatched areas indicate differences that exceed inter-annual variability. In (a), (c) and (d), gray areas indicate regions without observations. At least 0.5 kg m$^{-2}$ of liquid water in a grid cell in RACMO is required to be considered a day with liquid water.

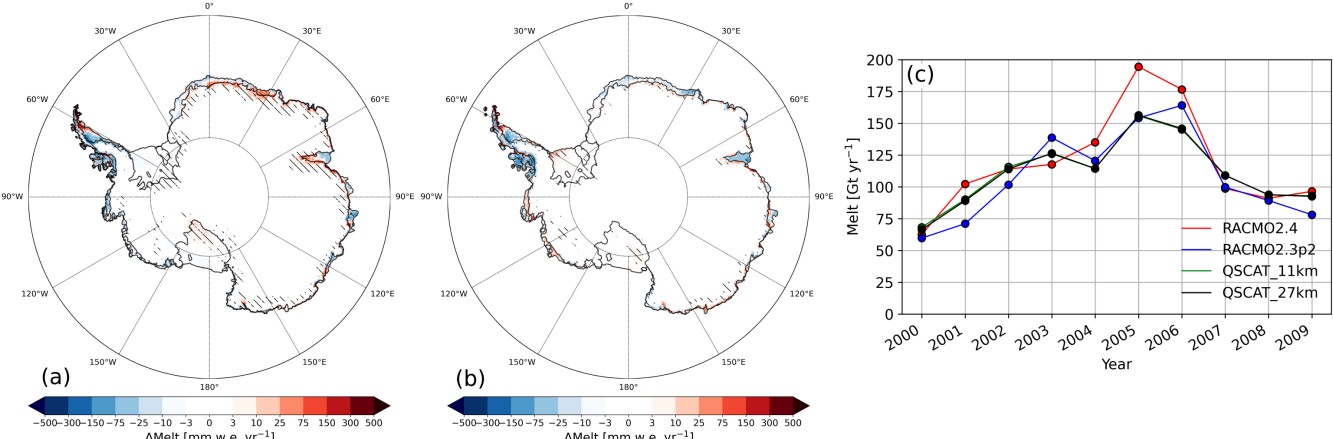

**Figure 7.** Annual accumulated melt with respect to the QSCAT melt product regridded to the RACMO grid for **(a)** R24 and **(b)** R23p2 (2000-2009). Hatched areas indicate differences that exceed inter-annual variability. **(c)** Time series of domain-integrated melt in Gt yr$^{-1}$ for R24, R23p2 and QSCAT regridded to the 11 km grid of R24 and 27 km grid of R23p2, with e.g. 2000 referring to the ablation season of 1999-2000.

## 4.2 Melt comparison with QSCAT

The liquid water observations of Picard (2022) based on SSM/I, AMSR-E and AMSR2 observations are unable to provide melt rates (Picard et al., 2022), and therefore cannot be used to evaluate the melt rate in RACMO. QSCAT, on the other hand, provides a calibrated seasonal total melt estimate (Trusel et al., 2013), allowing for a direct comparison of modeled seasonal melt water fluxes.

Figures 7a and 7b show the difference between modeled melt in R24 and QSCAT, and R23p2 and QSCAT, respectively. In the AP, melt fluxes are underestimated in both R24 and R23p2, except on the George VI ice shelf and north-east AP. Melt is somewhat underestimated on the Wilkins ice shelf, although this melt difference is below inter-annual variability, which is in contrast to the overestimated number of melt days shown in Fig. 6. This illustrates that too many wet days does not necessarily mean that too much melt is modeled. In West Antarctica, the melt flux in R24 is underestimated on the Abbot ice shelf, while it is overestimated in R23p2. For the ice shelves in East Antarctica, melt has increased in R24, in particular at the Amery shelf, removing the negative bias of R23p2, which is similar to Fig. 6. For the King Baudoin ice shelf and near the grounding line in DML, melt is now overestimated. Small amounts of melt are modeled in R24 near coastal zones in the EAIS that are absent in QSCAT.

Figure 7c shows a time series of melt integrated over the ice sheet for R24 and R23p2, and QSCAT regridded to the 11 km horizontal grid of R24 and 27 km horizontal grid of R23p2. Modeled melt in R24 is often in good agreement to the QSCAT melt product, in particular for 2000-2004 and 2007-2009, where melt in R23p2 is often too low. Albedo changes have led to considerably more melt in 2005 and 2006. A case study for melt in 2005 is discussed in Sect. 6.1.

### 4.3 Changes compared to R23p2

Melt in R24 is comparable to R23p2 when integrated over the ice sheet for 1979-2023 (Table 2). Regionally, differences are more considerable, especially over ice shelves (Fig. 4d). More specifically, melt has decreased on the Abbot, Pine Island, Thwaites and Getz ice shelves, while melt has increased on the Wilkins and George VI ice shelves in the AP. The George VI ice shelf is surrounded by steep slopes, and increased horizontal resolution in R24 results in a better representation of local topography. A better separation of the low-lying ice shelf surface from the steep valley cliffs leads to increased melt in R24. The melt changes are for many locations similar to those reported by Van Dalum et al. (2022). Radiative transfer in snow and ice is especially important on the ice shelves in DML and the Amery ice shelf in East Antarctica, as melt water often refreezes below the surface, locally enlarging snow grains and subsequently enhancing subsurface radiation absorption. Hence, the updated albedo scheme, including radiation penetration in snow and ice, is the primary cause of the altered melt patterns. This is investigated in more detail in Sect. 6.1. Modeled runoff remains negligible around the AIS, as nearly all melt water refreezes locally, except for a few grid points at the Amery and George VI ice shelves and the tip of the AP. Melt has also changed for several grid points at the Amery ice shelf, where bare ice is exposed in summer, for which a new albedo parameterization is used since R23p3.

## 5 Near-surface climate

### 5.1 Temperature comparison with observations

R24 compares well with near-surface AntAWS observations covering 1980-2021 (Wang et al., 2023) (Fig. 8a), especially considering the large geographical variety of AWS locations (Fig. 1). The fit line is close to the 1-on-1 line and the determination coefficient is high (0.93). Figure 8b shows that the modeled T2m follows the observed values closely with elevation. The temperature is somewhat too low in R24 (panel 2, Fig. 8b), which is rather typical for RCMs (Carter et al., 2022). The temperature is in particular lower for AWSs between 1000 and 2500 m elevation. The bias is almost 0°C for elevations typical for the East Antarctic Plateau, i.e., between 2500 and 3500 m, which is a considerable improvement compared to R23p2 (panel 3, Fig. 8b). For the other elevations, however, the bias in R23p2 is smaller. The T2m of R24 is within measurement uncertainties of observations for the two highest elevation bins. Figure 8c shows the RMSE as a function of measured T2m for R24 and R23p2. It shows that the RMSE has improved in R24 compared to R23p2 for almost all temperature bins and the spread is now lower, in particular for low temperatures (between -70 and -30°C).

With respect to IMAU AWS observations, T2m has a negative bias (Fig. 9h, -2.13 °C). A larger spread occurs between -30 and 0°C. Most of the T2m bias originates from AWS 16 located in DML, which is characterized by rough and inhomogeneous terrain, where R24 underestimates T2m considerably (bias of -6.92°C). If AWS 16 is left out, the T2m bias reduces to -1.35°C.

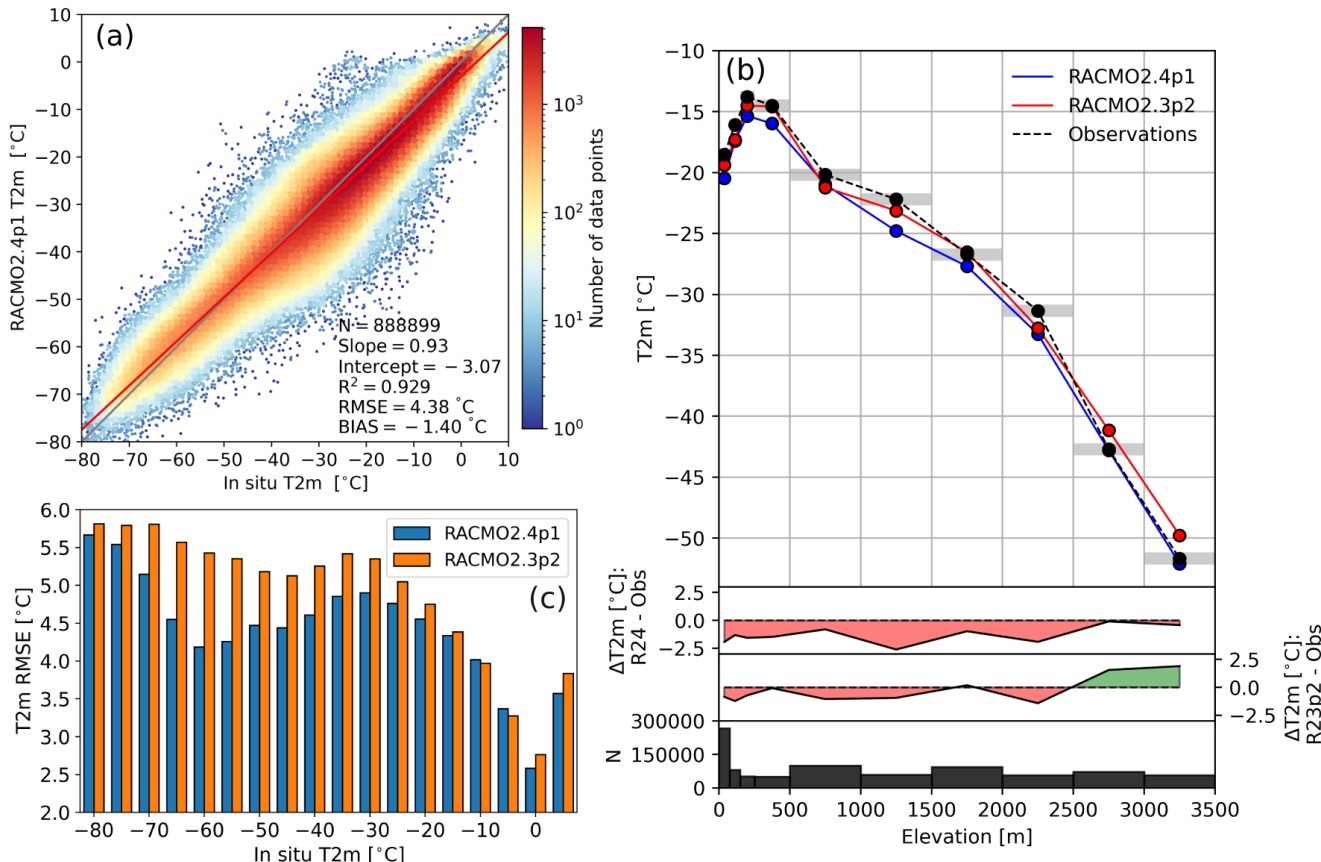

**Figure 8.** Daily-average 2-m temperature (T2m) for **(a)** R24 with respect to AntAWS observations (1980-2021). The bias, root-mean-square error (RMSE), determination coefficient ($R^2$), the total number of observations (N), the 1-on-1 fit line (black line) and the slope and intercept of orthogonal total least squares regression (red line) are also shown. **(b)** T2m as a function of elevation for R24, R23p2 and AntAWS observations for bins between 0, 75, 150, 250, 500, 1000, 1500, 2000, 2500, 3000 and 3500 m. Grey bins indicate the uncertainty of the measurements. Lower panels show per bin the T2m difference of R24 with observations, of R23p2 with observations, and N. **(c)** T2m RMSE with respect to AntAWS with 5 °C bins for R24 and R23p2. Locations of AntAWS are shown in Fig. 1.

## 5.2 Surface energy balance compared to observations

Compared to the IMAU AWS observations between 1995 and 2022 (Fig. 9 and Table 3, locations shown in Fig. 1), the $SW_d$ (Fig. 9a) is modeled well in R24, with a high determination coefficient (0.99). The RMSE is small (26.8 W m$^{-2}$), but the spread is larger between 300 and 400 W m$^{-2}$. The bias is positive with 8.5 W m$^{-2}$. A similar bias is shown for $SW_u$ (Fig. 9b), illustrating that the albedo is modeled well (discussed in Sect. 5.4). $LW_d$ (Fig. 9d) is underestimated (bias of -20.4 W m$^{-2}$), especially for low values when cloud cover has a limited impact. When considering only clear-sky observations, the bias is -25.4 W m$^{-2}$. MARv3.11 shows a similar bias as R24, with a mean bias of -20.4 W m$^{-2}$ without drifting snow and -14.9 W

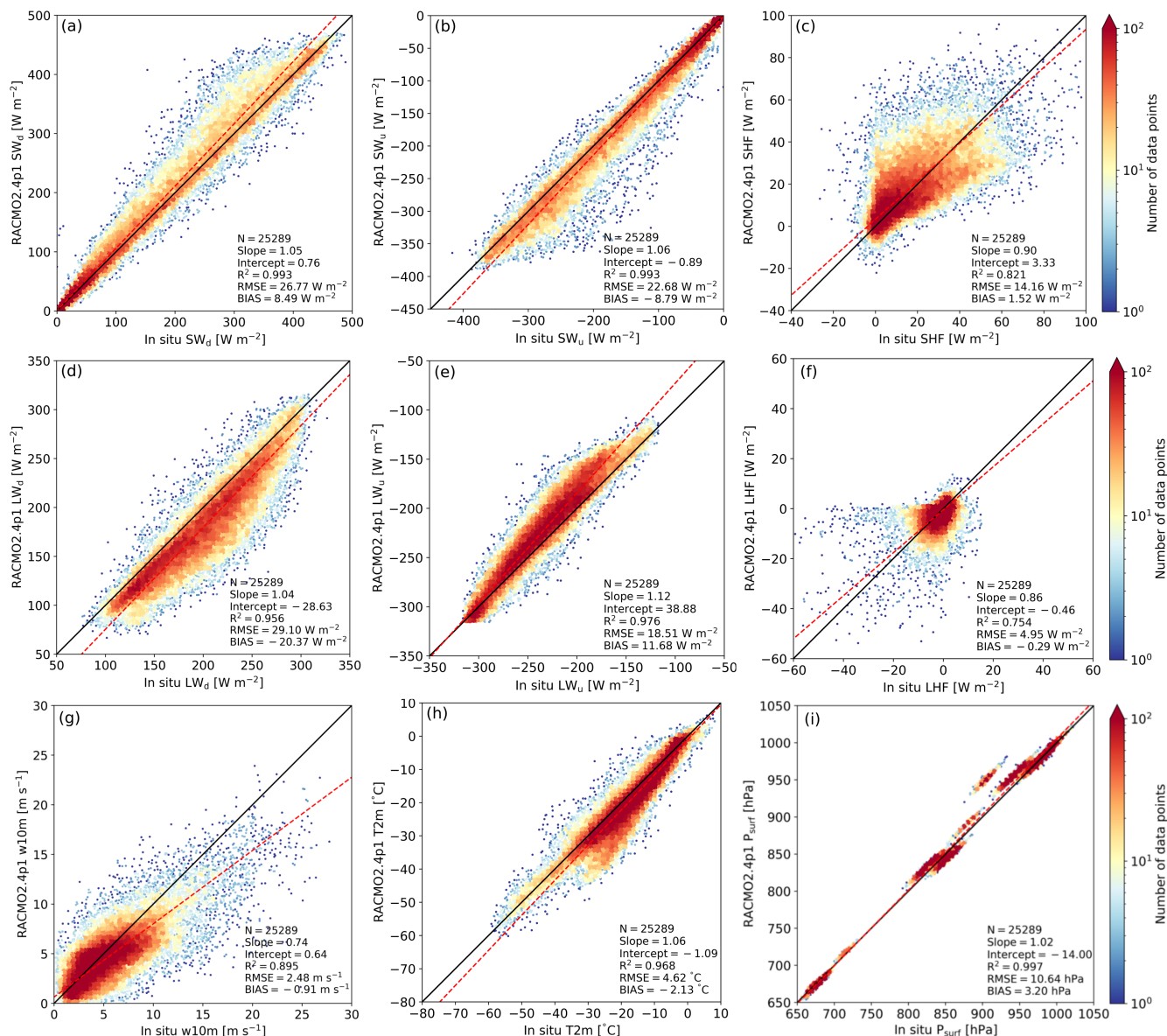

**Figure 9.** Daily-average SEB components for R24 with respect to the IMAU AWS observations for 1995-2022 for **(a)** $SW_d$, **(b)** $SW_u$, **(c)** SHF, **(d)** $LW_d$, **(e)** $LW_u$ and **(f)** LHF, and **(g)** 10-m wind speed (w10m), **(h)** 2-m temperature (T2m) and **(i)** surface pressure ($P_{surf}$). The bias, root-mean-square error (RMSE), determination coefficient ($R^2$), the total number of observations (N), the 1-on-1 fit line (black line) and the slope and intercept of orthogonal total least squares regression (red line) are also shown. Locations of IMAU AWSs are shown in Fig. 1.

m$^{-2}$ with drifting snow with respect to observations attached to a meteorological mast close to the coast of Adelie Land in east
420 Antarctica (Le Toumelin et al., 2021). $LW_u$ (Fig. 9e) is underestimated in R24 (bias of 11.7 W m$^{-2}$, note that $LW_u$ is negative), following a similar pattern as T2m (Fig. 9h). The sensible heat flux (Fig. 9c) is somewhat overestimated (1.5 W m$^{-2}$), despite

**Table 3.** Root-means-square error (RMSE), bias and determination coefficient ($R^2$) of daily-averaged SEB components, 10-m wind speed (w10m), 2-m temperature (T2m), surface pressure ($P_{surf}$) and surface albedo for R24 and R23p2 with respect to IMAU AWS observations (1995-2022). Locations are shown in Fig. 1. AWS 7 is not considered for the albedo.

| | | RMSE | | Bias | | $R^2$ | |
|---|---|---|---|---|---|---|---|
| Variable | Unit | R24 | R23p2 | R24 | R23p2 | R24 | R23p2 |
| $LW_d$ | $W\,m^{-2}$ | 29.1 | 22.9 | -20.4 | -6.2 | 0.96 | 0.96 |
| $LW_u$ | $W\,m^{-2}$ | 18.5 | 14.4 | 11.7 | -0.2 | 0.98 | 0.97 |
| $SW_d$ | $W\,m^{-2}$ | 26.8 | 26.8 | 8.5 | 6.1 | 0.99 | 0.99 |
| $SW_u$ | $W\,m^{-2}$ | 22.7 | 23.1 | -8.8 | -7.4 | 0.99 | 0.99 |
| LHF | $W\,m^{-2}$ | 5.0 | 5.9 | -0.3 | -1.4 | 0.75 | 0.68 |
| SHF | $W\,m^{-2}$ | 14.2 | 12.6 | 1.5 | 1.3 | 0.82 | 0.87 |
| w10m | $m\,s^{-1}$ | 2.48 | 2.50 | -0.91 | -0.50 | 0.90 | 0.88 |
| T2m | $°C$ | 4.62 | 4.55 | -2.13 | -0.01 | 0.97 | 0.96 |
| $P_{surf}$ | hPa | 10.6 | 12.3 | 3.2 | -7.9 | 1.00 | 1.00 |
| Albedo | - | 0.05 | 0.05 | 0.006 | 0.01 | 0.80 | 0.79 |

the underestimated wind speed for these stations (Fig. 9g). The bias of the latent heat flux (Fig. 9c) is small (-0.3 $W\,m^{-2}$). The spread, however, is considerable in observations and model output, showing that both observations and modeled turbulence can still be improved. Surface pressure is modeled well (Fig. 9i) and is only overestimated for AWS 11 in R24, where the elevation is too low (302 m in R24 while the AWS is located near the top of an ice rise at 700 m).

The RMSE, $R^2$ and bias of the SEB components, in addition to T2m, w10m and $P_{surf}$ for R24 and R23p2 are summarized in Table 3. The statistics of the shortwave radiative fluxes, turbulent fluxes and wind speed of R24 are comparable to R23p2, but the bias and RMSE of the longwave radiative fluxes have worsened. T2m is now underestimated. The bias and RMSE have improved in R24 for $P_{surf}$. Figure A2 shows the comparison with IMAU AWS in more detail for R23p2.

In future work, clouds and the impact on longwave radiation will be investigated by using the recently-launched EarthCARE satellite (Wehr et al., 2023), which may help identify mismatches in cloud location, structure, height, composition or optical properties. For clear-sky conditions, R24 determines longwave radiation by calculating absorption of all major gases, such as water vapour, $O_2$ and $CO_2$ on sixteen spectral bands for each vertical layer using the layer's transmittance and temperature (Mlawer et al., 1997). Modeling gas and aerosol concentrations correctly are thus essential. Future work will therefore involve investigating the impact of CAMS aerosols and SOCRATES cloud optical properties, which are are embedded in R24, on longwave radiation and how well they are tuned for the polar regions. For turbulence, a turbulent kinetic energy (TKE) closure scheme can potentially improve modeled turbulent fluxes, as the introduction of TKE as a prognostic variable that can move by advection can be used to simulate eddy diffusivity, allowing non-local effects to impact turbulence (Lenderink and Holtslag, 2000, 2004).

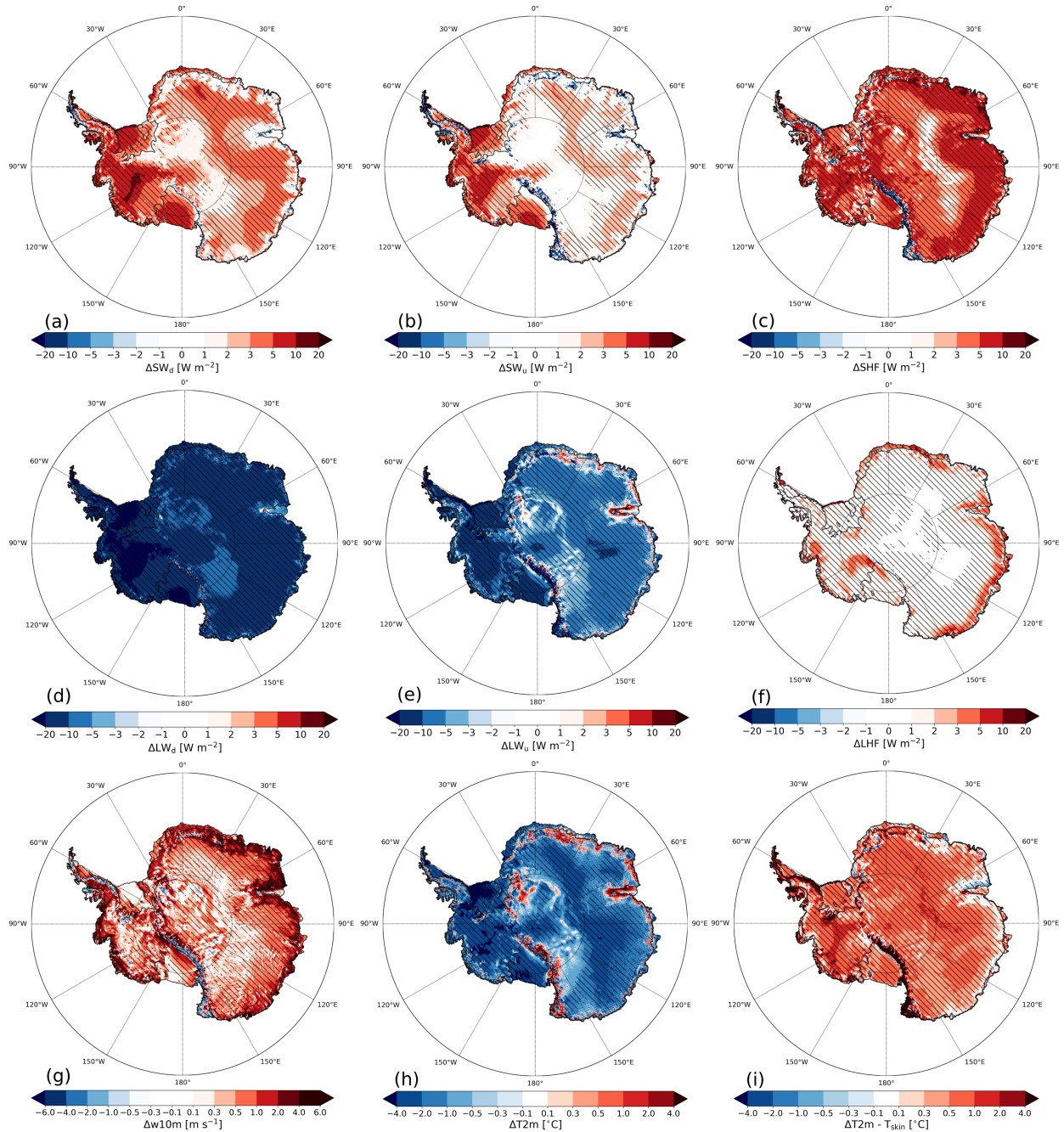

**Figure 10.** Yearly-average SEB difference (R24 − R23p2) for 1979-2023 for **(a)** SW$_d$, **(b)** SW$_u$, **(c)** SHF, **(d)** LW$_d$, **(e)** LW$_u$, **(f)** LHF, and **(g)** 10-m wind speed (w10m), **(h)** 2-m temperature (T2m) and **(i)** near-surface temperature gradient (T2m − skin temperature (T$_{skin}$)). Hatched areas indicate differences that exceed inter-annual variability.

## 5.3 Near-surface climate compared to R23p2

Differences of surface energy balance components for 1979-2023 of R24 with respect to R23p2 are shown in Fig. 10. Figure 10a shows that $SW_d$ has increased in R24, in particular in the WAIS. $SW_u$ (Fig. 10b) has also increased and follows the same pattern, but is not as strong as $SW_d$. The albedo has therefore changed as well, which is discussed in Sect. 5.4. In mountainous grid cells with a fractional ice mask (Fig. 1), $SW_u$ has decreased due to the exposure of bare soil during the ablation season, reducing the albedo. $LW_d$ (Fig. 10d) has decreased significantly over the entire AIS, and in particular in the WAIS, and coincides roughly with $SW_d$ changes. This illustrates that cloud cover and the cloud radiative impact on the surface has changed considerably in R24. The $LW_d$ decrease consequently leads to lower temperatures (Fig. 10h) and results in reduced $LW_u$. Increased wind speeds (Fig. 10g), in particular around the margins of the EAIS, and stronger near-surface temperature gradients (Fig. 10i) have prompted the turbulent fluxes to increase (SHF, Fig. 10c and LHF, Fig. 10f). Surface pressure ($P_{surf}$) differences are typically small (not shown), but are larger at the margins of the AIS, mainly in DML. This is likely caused by updated orography, inducing pressure differences.

## 5.4 Albedo

The yearly-average total-sky albedo of R24, determined as the ratio of the total annually-integrated incoming and outgoing shortwave radiative fluxes, is shown in Fig. 11a. RACMO models high albedo values ($>0.8$) almost everywhere on the AIS. In the WAIS, albedo values are higher due to persistent cloudy conditions and a high frequency of snow fall, resulting in frequent fresh snow at the surface. In the EAIS, bare ice is exposed in parts of the Amery ice shelf, resulting in a lower albedo. In-land of the Transantarctic Mountains, several grid cells have a fractional ice mask (See Fig. 1), with seasonal snow in winter and exposed rock in summer, resulting in a lower albedo.

The albedo of R24 compares well to IMAU AWS observations, as most data are close to the 1-on-1 line (Fig. 11c). The albedo of AWS 7 is often oddly low ($<0.75$), while fine-grained snow is almost always present here and high albedos ($>0.8$) are therefore expected (Pirazzini, 2004). Albedo observations of AWS 7 are thus excluded from the analysis. During some days, R24 calculates a high albedo typical for fresh snow (around 0.85), while measurements show a lower and varying albedo, which suggests that R24 models fresh snow that is not observed. On other occasions, such as for AWS 10, the albedo of R24 is too low, which is likely caused by cloud cover anomalies. Figure 11d shows the albedo for clear-sky conditions in R24 with respect to observations. The figure shows that the very high albedos ($>0.9$) do not occur for clear-sky conditions in R24, as one would expect (Kuipers Munneke et al., 2008). Occasionally, however, clear-sky conditions are modeled while cloudy conditions are observed, resulting in an underestimated albedo (shown in **A** in Fig. 11d). When both R24 and observations show cloudy conditions, the albedo is modeled well (shown in **B** in Fig. 11c). Compared to R23p2, the simulated total-sky albedo has improved in R24 (Table 3). While the RMSE remains similar to R23p2 (0.048 for R24 and 0.050 for R23p2), the bias (0.006 and 0.010) and $R^2$ (0.795 and 0.788) have improved.

The albedo of R24 is lower than R23p2 for most of the AIS (Fig. 11b), including on the Amery ice shelf where bare ice is exposed during summer for several grid points. This has several reasons. In R24, the spectral albedo model TARTES is

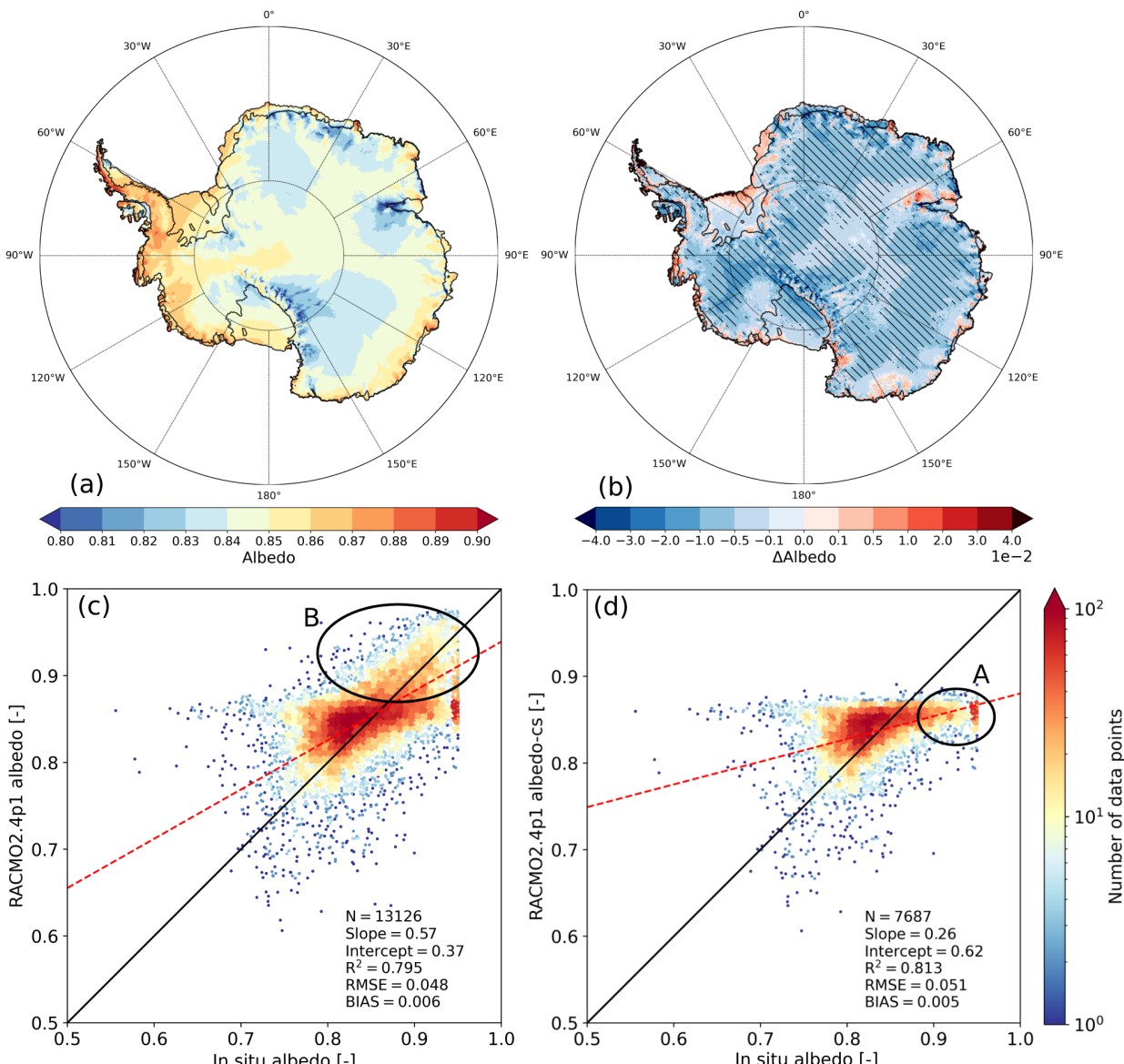

**Figure 11.** Yearly-average total-sky albedo for **(a)** R24 and **(b)** the difference between R24 and R23p2. Hatched areas indicate differences that exceed inter-annual variability. Daily-averaged albedo for R24 with respect to IMAU AWS observations (1995-2022) for **(c)** total-sky and **(d)** clear-sky conditions. A minimum of 100 W m$^{-2}$ of daily-averaged SW$_d$ is necessary for the day to be considered. Conditions are considered to be clear-sky when the ice water and liquid water path are below 0.01 kg m$^{-2}$. The bias, root-mean-square error (RMSE), determination coefficient ($R^2$), the total number of observations (N), the 1-on-1 fit line (black line) and the slope and intercept of orthogonal total least squares regression (red line) are also shown. **A** and **B** are discussed in the text. Locations of IMAU AWSs are shown in Fig. 1. Observations from AWS 7 are not considered in this analysis.

incorporated using SNOWBAL, which also takes changes in the spectral distribution of irradiance into account. This results in an albedo decrease with high SZA, as the solar radiation consists of relatively more infra-red (IR) radiation, for which the albedo is lower (Van Dalum et al., 2019). With cloud cover, however, the albedo increases, as IR radiation is filtered out and radiation that reaches the surface consists mostly of visible light, for which the albedo is high (Gardner and Sharp, 2010). Furthermore, radiative transfer typically lowers the albedo, as subsurface layers with larger grain radii can now impact the albedo. As discussed in Van Dalum et al. (2022), incorporating TARTES in RACMO revealed inaccuracies in the subsurface grain size evolution existing in R23p2 that also impact the albedo, which have been corrected in R23p3 and are now included in R24.

## 6 Case studies

Two case studies are presented here to highlight the impact of several processes and changed parameterizations on model output. A year with exceptionally large melt fluxes, especially on the King Baudouin and Amery ice shelves, is studied to find the relevant processes involved. Another case study involves seasonal snow in the McMurdo Dry Valleys, which is one of the few locations in Antarctica where multiple grid cells have fractional ice cover and where the new multi-layer seasonal snow scheme can be evaluated.

### 6.1 Melt in the austral summer of 2005

Integrated over the ice sheet, R24 compares generally well with the QSCAT melt product, but for some years melt is overestimated (Fig. 7c). Especially 2005 is a year with extensive melt, where R24 deviates considerably from the QSCAT observations, particularly at the Amery and King Baudouin ice shelves (Fig. 12a). Here we investigate the origin and cause of these melt differences.

The melt difference patterns of R24 with respect to QSCAT (Fig. 12a) are similar to the 2000-2009 average (Fig. 7a), but the King Baudouin and Amery ice shelves stand out with increased melt in Fig. 12a. On a selected grid point on the east side of the Amery ice shelf, melt starts in December 2004 for both R24 and R23p2 (Fig. 12b). On the 25th of December, melt decreases in R23p2 and virtually no melt is modeled from the 2nd of January onward with the arrival of fresh snow. As no radiative transfer is calculated in R23p2, the albedo increases strongly even with a thin snowpack, inhibiting melt. In R24, however, melting continues until the fresh snow layers grow thick enough on the 28th of January. The occurrence of melt in January in R24 is in agreement with AMSR-E observations of liquid water (lower panel in Fig. 12b). The SEB components of R23p2 and R24 are similar to each other for most of the year on this grid point on the Amery ice shelf (Fig. 12b). Melt rates and SEB components are similar during the start of the ablation season in December, when the fresh snow layer is melting. After the fresh snow layer has melted away in January, however, $SW_u$ becomes lower in R24 with respect to R23p2 while $SW_d$ remains similar, consequently increasing $SW_{net}$. As the other fluxes in R24 remain similar to R23p2, cloud effects are limited and aforementioned albedo changes therefore lead to increased melt.

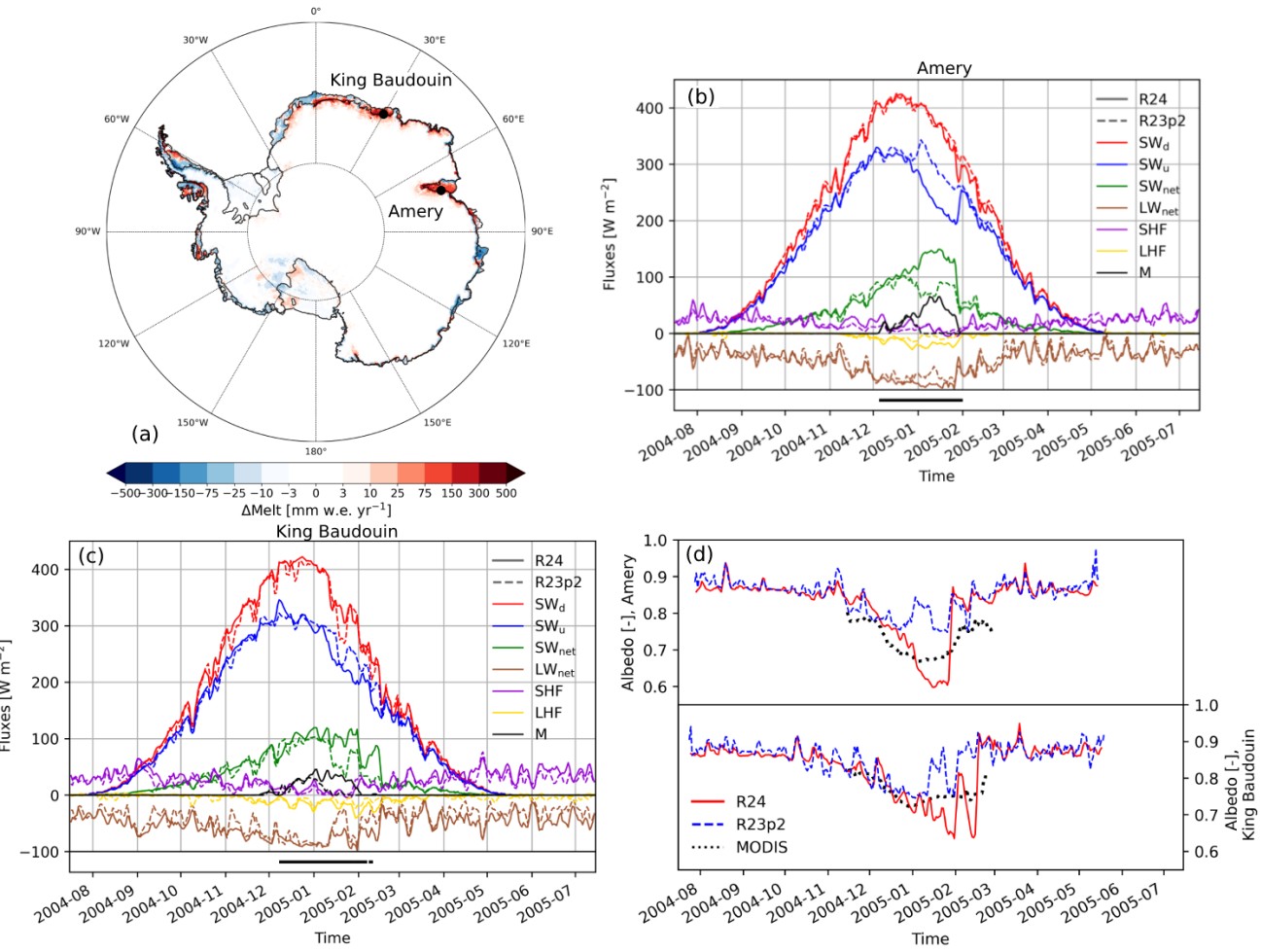

**Figure 12. (a)** Modeled melt with respect to the QSCAT melt product regridded to the RACMO grid for R24 for 2005. Time series of daily-average SEB components and melt (M, in W m$^{-2}$) in R24 and R23p2 for grid points on **(b)** the Amery ice shelf and **(c)** the King Baudouin ice shelf (locations shown in (a)). Days with liquid water observed by AMSR-E are shown as a black line in the panels below. **(d)** Time series of the broadband albedo of R24 and R23p2, which includes contributions of both direct and diffuse radiation, for the same grid points in the Amery ice shelf (upper panel) and King Baudouin ice shelf (lower panel). Observed MODIS white-sky albedo during summer is shown with a dotted line. No albedo is modeled during the polar night. For (b)-(d), a three-day running mean is applied to reduce scatter.

A similar signal is found for the King Baudouin ice shelf (Fig. 12c), where melting starts on the 24th of November and persists through January in R24, which is in agreement with AMSR-E liquid water observations (lower panel of Fig. 12c), while almost no melting occurs in January in R23p2. Apart from roughly a week with more cloud cover in mid January in R23p2 (from the 16th to the 24th of January), the SEB components of R24 are also similar to R23p2 in January, except for SW$_u$. The albedo is therefore also the leading factor for increased melt at this location.

Figure 12d shows the summer 2004-2005 albedo of both RACMO versions and MODIS white-sky albedo for the Amery ice shelf (upper panel) and King Baudouin ice shelf (lower panel). During the accumulation season and the start of the ablation season, the albedo of R24 and R23p2 are similar. As the ablation season progresses, the surface snowpack becomes saturated with melt water that refreezes overnight. Refreezing of melt water results in rapid snow grain size growth and more absorption of SW radiation, lowering the albedo (Gardner and Sharp, 2010). As a radiative transfer scheme is implemented (Van Dalum et al., 2019, 2022) and the refreezing grain size is larger in R24, the albedo during melt days becomes lower in R24 than R23p2. This effect is particularly strong on the Amery ice shelf, where a fresh snowpack of only several decimeters forms in the accumulation season that melts and refreezes in summer. At the King Baudouin ice shelf, the effect of increased refreezing grain size is also strong in 2005, as the amount of snowfall is low in the three months preceding the ablation season, resulting in an unusually thin snowpack at the onset of the melt season. The albedo fully recovers with the onset of the accumulation season in February, when fresh snow covers refrozen snow. In R24, a short melt event at the King Baudouin ice shelf in February delays the albedo recovery by several days.

Compared to the MODIS white-sky albedo, the R24 albedo follows the decline in December, but continues for too long, resulting in an underestimated albedo in January. Only a weak albedo decline is calculated by R23p2 and the albedo is too high during the ablation season (around 0.8). For the King Baudouin ice shelf, R24 and R23p2 capture the gradual albedo decline in December well, but the albedo is also too high in January for R23p2 and too low for R24. The albedo has somewhat improved in summer in R24, resulting in more melt, but is often too low. Note, however, that for RACMO both direct and diffuse radiation are used to calculate the albedo, while MODIS white-sky albedo is determined in the absence of a direct component. This case study shows the importance of the albedo feedback on melt, and in particular the refreezing grain size that is a key parameter in modeling the snow melt-albedo feedback.

## 6.2 Seasonal snow in the McMurdo Dry Valleys

The McMurdo Dry Valleys are one of the few locations with ice-free valleys in Antarctica, and several grid points in R24 are therefore completely or partially ice-free (Fig. 1). A seasonal snow cover forms during the accumulation season and melts during the ablation season in ice-free areas in the McMurdo Dry Valleys. As these areas are ice-free in the present-day climate, any snow cover that forms in R24 should disappear at the end of the ablation season. The McMurdo Dry Valleys, however, are characterized by low temperatures and small amounts of precipitation, and only a small overestimation of precipitation could easily lead to too much snow that does not completely disappear during the ablation season.

Figure 13a shows the snow depth as a function of glaciated tile fraction at the end of the ablation season for several years. The figure shows that virtually all snow melts for grid points with no permanent ice (a glaciated tile fraction smaller than 0.1). Despite that most snow melts for grid points with a small glacier fraction, i.e., between 0.1-0.2, the mean snow depth is increasing over time (colored lines) and snow therefore gradually accumulates. Snow depth varies considerably between grid cells within this bin, as virtually all snow melts for some grid cells, while a thick snowpack (>5 m) forms for others. The upper and lower end of the grey bar plots show the extreme of each bin. While some grid cells are ice-free or partially covered by snow at the end of the ablation season in 2022 for the 0.1-0.2 and 0.2-0.3 bins (Fig. 13b), all grid cells are fully covered

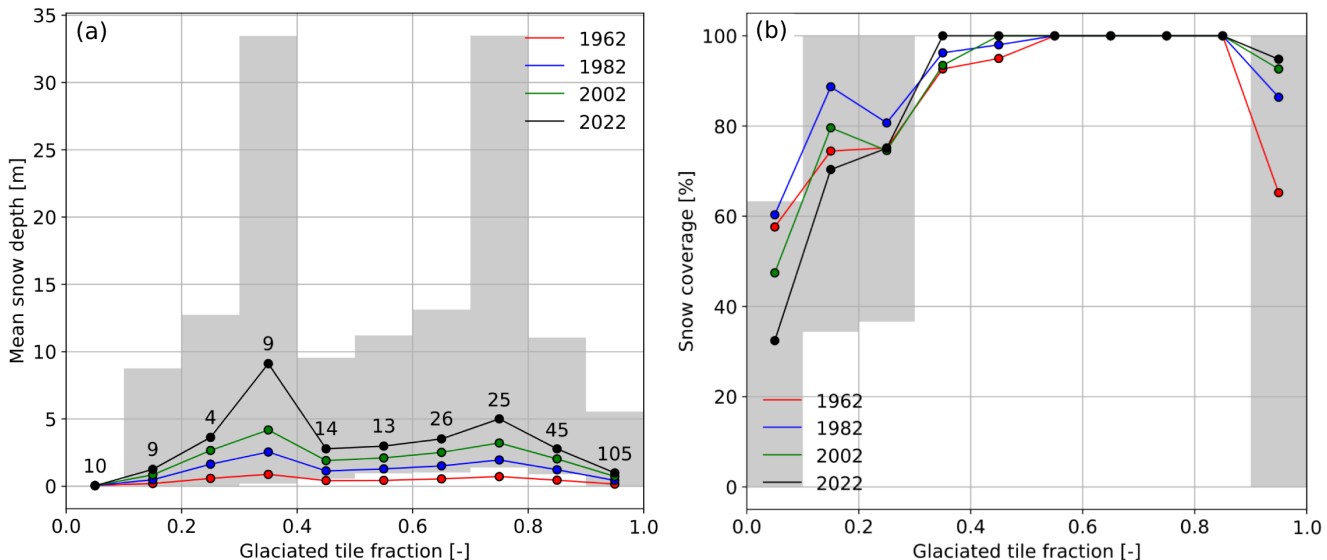

**Figure 13. (a)** Mean snow depth and **(b)** snow coverage in R24 on ice-free tiles as a function of glaciated tile fraction at the end of the ablation season (April the first) for several years for the McMurdy Dry Valleys (all considered grid points are shown in Fig. 1). Grey bars show the minimum and maximum snow depth at 01-04-2022 for each glaciated tile fraction bin (with a bin size of 0.1). Numbers above the colored lines in (a) indicate the number of glaciated grid points available for each bin. No permanent ice is modeled for the 0.0-0.1 glaciated tile fraction bin.

between 0.3 and 0.9. Snow coverage is only 0 at the end of the ablation season for a few grid points with no permanent ice (0.0-0.1 bin) and with a very high fraction (0.9-1.0 bin). The low snow coverage for some of the grid cells of the 0.9-1.0 bin is

545    likely due to low amounts of precipitation, as they are located at a higher elevation and further inland in the dry interior of the EAIS. Some inter-annual variability is visible for the lower fractions (0.0-0.3).

    Figure 13 illustrates that the approach of splitting a grid point into several tiles to represent subgrid variability has its shortcomings. In RACMO, the SEB components and skin temperature are calculated for each tile and combined into a grid-cell weighted average. There is, however, only one atmosphere for each grid cell, which is used for all tiles. As temperatures are

550    typically lower on the ice sheet than on ice-free tiles, the ice sheet tends to lower the near-surface atmosphere temperature. The glaciated surface tile therefore impacts the ice-free tile with seasonal snow by lowering the atmospheric temperature and by reducing melt. This effect strengthens with increased glaciated tile fraction, therefore explaining the patterns shown in Figure 13.

## 7    Summary and conclusions

555      – We presented a new near-surface climate and SMB product for the Antarctic ice sheet using the new model version RACMO2.4p1 (R24).

- Integrated over the ice sheet, including ice shelves, the modeled SMB is 2546 Gt yr$^{-1}$, with an interannual variability of 133 Gt yr$^{-1}$.

- Compared to RACMO2.3p2, the SMB in the WAIS and the AP have improved due to precipitation and sublimation changes caused by the advection of snow hydrometeors and alterations in the blowing snow parameterization.

- Melt events are modeled around the AIS and compare well with remote sensing observations.

- Compared to AWS observations, shortwave radiative fluxes, albedo and near-surface temperature are modeled well, but further research is needed to improve downward longwave radiation and turbulence.

- Two case studies are presented that show the impact of albedo and radiative transfer on melt and explore the impact of the introduction of a fractional glacier mask.

For the near-surface climate and SMB product presented in this study, R24 is run on an 11 km horizontal resolution grid that includes Antarctica, the Drake Passage and the southern tip of South America for 1960-2023. We evaluated the model results with in-situ and remote sensing observations, and compared it to the previous operational RACMO version, RACMO2.3p2, which is run on a 27 km horizontal resolution grid for 1979-2023. With respect to RACMO2.3p2, several aspects of the model are updated in R24, including an update to the IFS physics module, a new multi-layer snow scheme for seasonal snow, an updated fractional ice mask and a new albedo and radiative transfer in snow scheme.

Comparing the modeled SMB in R24 to the AntSMB data set shows a low bias, with -28.3 mm w.e. yr$^{-1}$, but the SMB remains somewhat underestimated for high-accumulation areas. On the EAIS, the SMB has decreased due to lowered precipitation and more sublimation, the latter is caused by adjustments in the blowing snow parameterization. At the margins of the EAIS, an alternating pattern of increased and decreased precipitation occurs due to newly implemented advection of snow hydrometeors. A considerable SMB increase is seen in the AP, where the advection of snow hydrometeors delivers larger quantities of snow to the ice sheet. The SMB has improved in several drainage basins, in particular in the WAIS, but the bias is still large for others, especially in the AP. Integrated over the ice sheet, the SMB is 2546 ± 133 Gt yr$^{-1}$. The melt flux is small (124 ± 31 Gt yr$^{-1}$), but is considerable on several ice shelves. All SMB components are comparable to R23p2, except sublimation, which has increased by 61 Gt yr$^{-1}$.

Days with liquid water present in the snowpack can be considerable on ice shelves. The number of wet days agrees well when compared to remotely-sensed passive microwave radiometer observations and differences are often insignificant. Some areas in the AP, however, have aquifers present in R24 at least a meter below the surface that cannot be observed by the satellite sensors used here. Melt fluxes are compared to the calibrated seasonal QuikSCAT satellite product, which shows that melt amounts are often modeled well, and differences do not exceed 70 Gt yr$^{-1}$, except for 2005, which is studied in more detail. On the Wilkins and Larsen C ice shelves, however, not enough surface melt is modeled. Increased melt on ice shelves of the EAIS is in agreement with observations, except for the Amery and King Baudouin ice shelves, where it is overestimated for some years. For virtually all locations, runoff is negligible and nearly all melt water refreezes locally according to RACMO.

The SEB components are compared to IMAU AWS observations that allow closure of the SEB, including melt rates. Short-wave radiative fluxes compare well to observations (bias of 8.5 and -8.8 W m$^{-2}$ for downward and upward shortwave radiation, respectively), but differences for longwave radiative fluxes are larger (bias of -20.4 and 11.7 W m$^{-2}$ for downward and upward longwave radiation, respectively). In particular the downward longwave radiation is underestimated during both clear-sky and cloudy conditions. Biases of modeled turbulent fluxes are typically small (-0.3 and 1.5 W m$^{-2}$ for the latent and sensible heat flux, respectively), but the spread is large (RMSE of 5.0 and 14.2 W m$^{-2}$, respectively). The addition of a spectral snow albedo and radiative transfer scheme has resulted in a lower albedo on the AIS with respect to R23p2, and shows improvements compared to observations. Near-surface temperature is modeled well when compared to AntAWS observations, with a bias of -1.40 and a RMSE of 4.38 °C, in particular at the higher-elevated areas of the EAIS. Further research is needed to improve the downward longwave radiation and turbulent fluxes.

Two case studies are presented to highlight the impact of processes and parameterizations on model output. The first case study involves a strong melt year on the King Baudouin and Amery ice shelves, showing that the albedo parameterization has a considerable impact on melt. Increased refreezing grain size and the addition of a spectral albedo and radiative transfer scheme lowers the albedo and can extend the melting season by up to a month for these locations, bringing it in agreement with satellite observations. The introduction of a new fractional glacier-ice mask in R24 results in grid cells that are partly covered by glacial ice or are completely ice-free. A case study of seasonal snow in the McMurdo Dry Valleys reveals that R24 is able to melt most seasonal snow on ice-free grid cells at the end of the ablation season, but has trouble melting snow on the ice-free fractions of grid cells that are also partially covered by glacial ice. In future work, R24 will be used for Greenland and to study future climate storylines for Antarctica.

## Appendix A: Figures

*Author contributions.* Writing of the manuscript, model development and analysis of model output are led by CTvD. MvT processed the IMAU AWS data and provided support. All authors contributed to discussions on the results and the manuscript.

*Data availability.* Monthly RACMO2.4p1 data are available for the SMB components, SEB components and near-surface variables like temperature, wind speed and pressure. The data can be accessed here: https://doi.org/10.5281/zenodo.14217231 (Van Dalum et al., 2024a). RACMO2.3p2 from Van Wessem et al. (2023a) are available here: https://doi.org/10.5281/zenodo.7760490. The AntSMB data are available from Wang et al. (2021) (https://doi.org/10.11888/Glacio.tpdc.271148). AntAWS data are available from Wang et al. (2023) (https://doi.org/10.48567/key7-ch19). IMAU AWS data are uploaded to PANGAEA: https://doi.pangaea.de/10.1594/PANGAEA.974080 (Van Tiggelen et al., 2025). QuikSCAT data are described by Trusel et al. (2013). Passive microwave radiometer observations of liquid water in snow made by Picard (2022) are available here: https://doi.org/10.18709/perscido.2022.09.ds376.

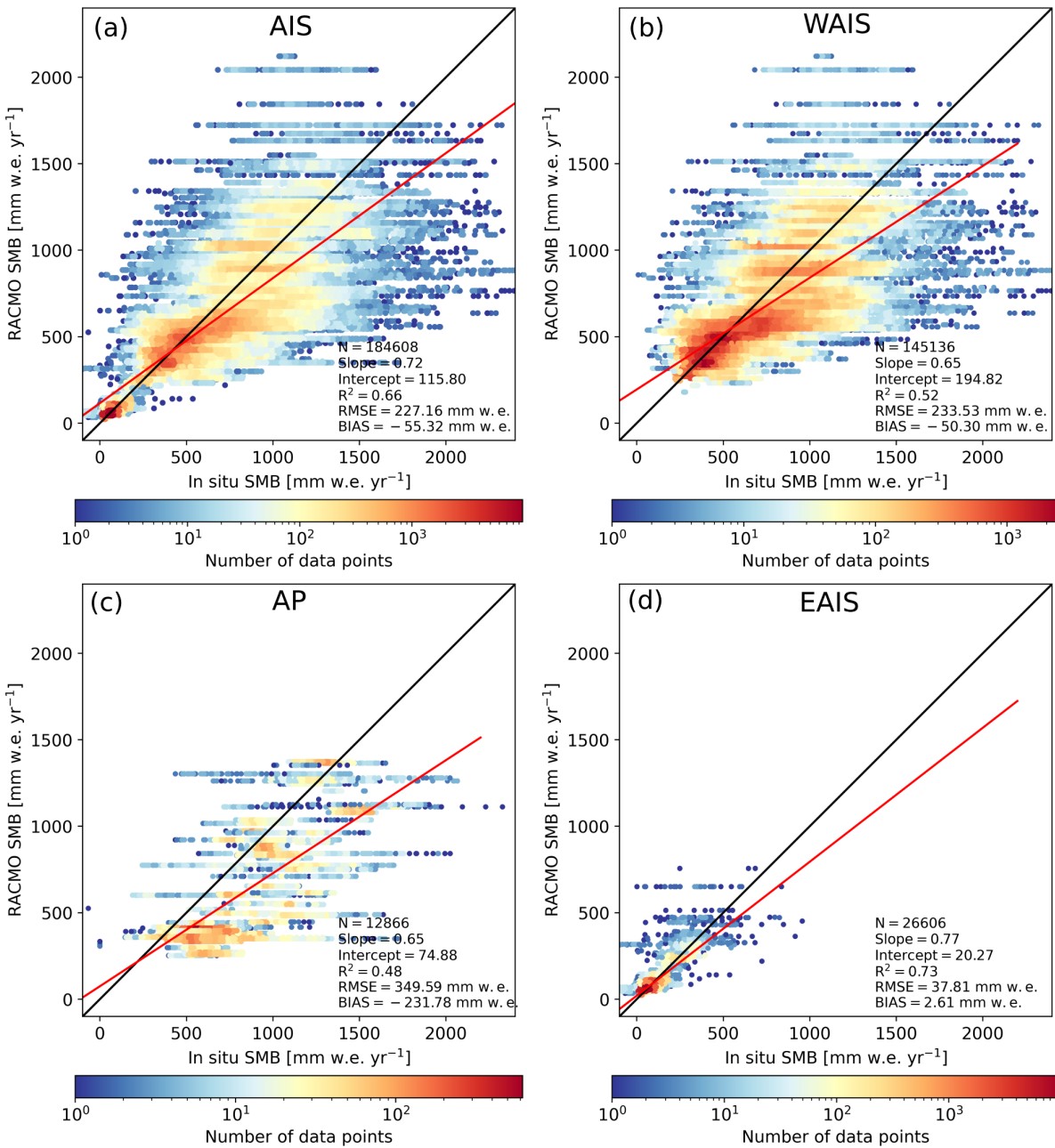

**Figure A1.** Surface mass balance of R23p2 with respect to observations of the AntSMB data set for 1979-2017 for **(a)** the entire AIS, **(b)** the WAIS, **(c)** the AP and **(d)** the EAIS. The 1-on-1 line is shown in black and the linear regression fit in red. The number of observations (N), slope and intercept of the linear regression fit, determination coefficient ($R^2$), root-mean-square error (RMSE) and bias are also shown.

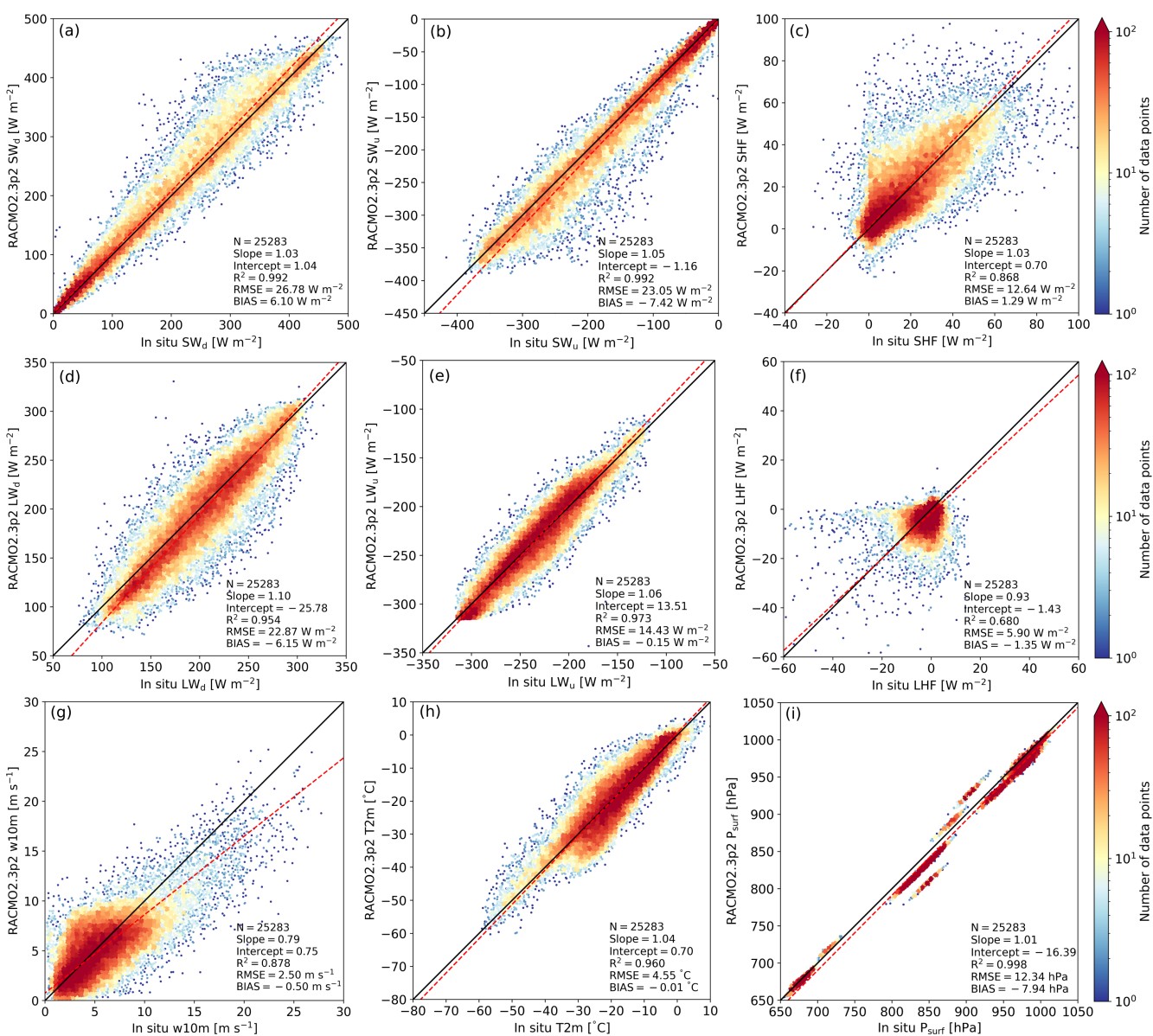

**Figure A2.** Daily-averaged surface energy balance components for R23p2 with respect to the IMAU AWS observations for 1995-2022 for **(a)** $SW_d$, **(b)** $SW_u$, **(c)** SHF, **(d)** $LW_d$, **(e)** $LW_u$ and **(f)** LHF, and **(g)** w10m, **(h)** T2m and **(i)** $P_{surf}$. The bias, root-mean-square error (RMSE), determination coefficient ($R^2$), the total number of observations (N), the 1-on-1 fit line (black line) and the slope and intercept of orthogonal total least squares regression (red line) are also shown. Locations are shown in Fig. 1.

*Competing interests.* At least one of the (co-)authors is a member of the editorial board of *The Cryosphere*. The authors have no other competing interests to declare.

*Acknowledgements.* This publication was supported by PolarRES. This project has received funding from the European Union's Horizon 2020 research and innovation programme under grant agreement No. 101003590 (PolarRES). We acknowledge the ECMWF for storage facilities and computational time on their supercomputer. We also acknowledge Ghislain Picard for providing feedback and technical support on the usage of his passive microwave radiometer observations product of liquid water in snow.

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
