# Peer review of "The surface mass balance and near-surface climate of the Antarctic ice sheet in RACMO2.4p1"

_EGUsphere, 2024_

## Author Response (AR1)

Referee comment responses on the manuscript:

***The surface mass balance and near-surface climate of the Antarctic ice sheet in RACMO2.4p1***

by C.T. van Dalum et al.

We would like to thank the referees for their comments and we address them here. In black the comment, in orange the response, in blue the changes that we would implement in the manuscript. All line numbers refer to the old manuscript version.

**Review #1, anonymous**

**Review of "The surface mass balance and near-surface climate of the Antarctic ice sheet in RACMO2.4p1" by Christiaan T. van Dalum et al.**

**The Cryosphere: egusphere-2024-3728**

**General comments:**

The authors introduce and evaluate the state-of-the-art version of the Regional Atmospheric Climate Model (RACMO), version 2.4p1 (R24), applied in Antarctica in this paper. Because RACMO is widely used in the cryosphere community to investigate ice sheet surface mass balance, knowing the current state of the state-of-the-art version of RACMO must be interesting for readers. Also, this paper is informative for readers interested in climate model development in polar regions. I have found that the model development is solid, and the evaluation processes are reliable. I would suggest that this paper can be published after revisions.

One major issue for me is composition of this paper (all the major specific comments are related to this point): The authors want to show comparison results between R24 and the previous version R23p2 before showing evaluation results of R24 with respect to in-situ measurements, which I could not fully understand. I think the model evaluation results with respect to in-situ measurements should always come first in the result section for this kind of model development paper. This is because, if the model performance with respect to in-situ measurements is not good, comparisons against the previous model versions and/or model data analyses have no meaning. Therefore, confirmation of the suitable model performance is the first priority for this kind of paper in my humble opinion. Please recall that the authors' research group always does like this when they perform SEB (surface energy balance) analyses, which I think is the solid scientific approach.

**Specific comments (major)**

Section 3: I believe the authors should start by presenting model evaluation results with respect to in situ measurements (in particular, Sect. 3.3 and Table 2) before showing interannual variations of model-simulated SMB and its components. The current format mixes evaluation results and analyses of interannual variations, which was a bit difficult for me to follow the content. Related to this point, in Sect. 3.1 (around L. 221 ~ 222), I wanted to know about the

impacts of enhanced horizontal resolution in R24, which was explained later in Sect. 3.3. If the current contents in Sect. 3.3 come before the current contents in Sect. 3.1, it will be easier for readers to understand the authors' thoughts on the model performance.

We had set up the paper in this order as the comparison with R23p2 gives first insight in the changes induced by this new model version. The subsequent evaluation against observation shows whether these changes are an improvement or not. However, we agree that presenting the evaluation first is the common order, and as the reviewer requests this, we will present the comparison with observations first in the revised manuscript. Therefore, we have moved Section 3.3, Figure 5 and Table 2 to the beginning of Section 3. The first paragraph of Sect. 3.1 is still the first paragraph of Section 3, and is thus merged with the AntSMB comparison section. Several minor changes are also implemented in the text to accommodate for these changes. More details about the minor changes will be available in the marked-up version.

Section 4: Like Sect. 3, I think the contents in Sects. 4.2 and 4.3 should be presented before that in Sect. 4.1. I think model evaluation with respect to in-situ measurements should always come first.

We have moved Section 4.1 to the end of Section 4, such that Section 4.2 and 4.3 are presented first. The only necessary change in the text is the first sentence of Section 4.1 (now Section 4.3):

Beginning of the new Section 4.1: Melt in R24 is comparable to R23p2 when integrated over the ice sheet for 1979-2023 (Table 1).

Section 5: The same as my comment on Sect. 4.

Based on your suggestions, we have restructured Section 5 extensively, such that the comparison with observations are discussed before the comparison with RACMO2.3p2. More specifically, Section 5.3 now becomes Section 5.1 and is renamed to 'Temperature comparison with observations' and Figure 11 becomes Figure 8. This section now includes the comparison with AntAWS observations and with IMAU AWS observations, with the latter moved from the Surface energy balance section (previously Section 5.1). The surface energy balance section is now split into two, becoming Section 5.2 and Section 5.3. Section 5.2 is called 'Surface energy balance compared to observations', where the various SEB components are analysed with respect to IMAU AWS observations. It now also includes a paragraph where potential improvements and future work is discussed regarding longwave radiative and turbulent fluxes (see also Review #2). Section 5.3 is now called 'Near-surface climate compared to R23p2', and includes the SEB components, wind speed and surface pressure comparison with R23p2. Figure 8 is moved to this section and is now Figure 10. Section 5.4 'Albedo' is also restructured and Figure 10 has become Figure 11. The first paragraph discusses how the albedo is modelled spatially in R24. The second paragraph compares RACMO with IMAU AWS observations, also for clear-sky conditions, and in the third and last paragraph R24 is compared to R23p2. More details about the changes will be available in the marked-up version.

**Specific comments (minor)**

L. 105 "Rain and snow are now also prognostic and separate variables, allowing them to have a fall speed and to be moved by advection of the ambient air flow.": Is it true? How did the authors calculate rainfall and snowfall amounts in RACMO2.3?

You are right that this is somewhat confusing. Rain and snow were calculated before in RACMO2.3 as separate entities within the cloud scheme of the model, but they would precipitate instantaneously to the ground. The difference with RACMO2.4 is that calculated rain and snow do not precipitate immediately to the ground but are retained as separate prognostic variables. Based on the fall speed and the time step used in the model, some rain and snow can be conserved in the atmospheric model levels, which can then be moved by the ambient flow before precipitating at a later stage.

L105: Rain and snow are now separate prognostic variables, allowing them to have a fall speed and to be moved by advection of the ambient air flow instead of precipitating instantaneously to the ground.

L. 121 (Sect. 2.1.3): Please indicate the time interval of the RACMO output data.

L121: In this study, R24 is run on an 11 km horizontal resolution grid for a domain that includes Antarctica, the Drake Passage and the southern tip of South America, in compliance with the Antarctic CORDEX domain (https://climate-cryosphere.org/antarctic-cordex/, last access: 14 May 2025) and daily RACMO output are used.

L. 217 ~ 218 "where exposed blue ice lowers the albedo (Tuckett et al., 2021).": Is it an observed result or what R24 simulates? Please clarify.

We have changed the following to clairify this:

L217: The only places with a (near-)zero SMB are in the western part of the Amery ice shelf near the grounding line, where exposed blue ice is modelled that lowers the albedo and where melt water ponds are frequently observed (Tuckett et al., 2021).

L. 251 ~ 252: The authors indicate that a change of 5.9 Gt yr-1 is insignificant and a change of 5.8 Gt yr-1 is significant, which sounds strange. Please reformulate.

This is indeed confusing but correct though, due to much larger interannual variability in melt and runoff compared to snow drift erosion. To avoid this confusion, we have changed it to the following:

L251: Melt and runoff changes are small (5.9 and 0.9 Gt yr-1, respectively), but by percentage runoff has increased considerably (22.0%). Blowing snow erosion has increased significantly due to higher wind speeds over the AIS (107.4%), but it still relatively small (11.2 Gt yr-1)

L. 340 ~ 342 "In the Wilkins ice shelf, these aquifers exist for several months before refreezing during winter, and are typically located at least 1 m below the surface, thus undetectable by the remote sensing technique used here.": Is it an observed result? If so, please add a reference for this argument.

These aquifers are located below 1 m in RACMO2.4 and refreeze during winter. Such aquifers are undetectable by the remote sensing methods used in our study.

L340: This is due to the presence of aquifers in R24 below the surface layers. In the Wilkins ice shelf, these aquifers exist for several months before refreezing during winter in R24, and

are typically located at least 1 m below the surface. Such aquifers are therefore undetectable by the remote sensing technique used here.

L525: Some areas in the AP, however, have aquifers present in R24 at least a meter below the surface that cannot be observed by the satellite sensors used here.

L. 466 (and L. 180) and Fig. 12: White-sky albedo is defined as albedo in the absence of a direct component when the diffuse component is isotropic (e.g., https://lpdaac.usgs.gov/products/mcd43d61v006/). What is the definition of the RACMO2 albedo here? Is it white-sky albedo? Please clarify.

RACMO is unfortunately unable to produce a white-sky albedo and there are no separate direct or diffuse components available to determine a white-sky or black-sky albedo. We deemed the white-sky MODIS albedo product to be the closest to what we have available in RACMO. We added a clarification where we note the difference between the products in the manuscript:

L187: The MODIS white-sky albedo is closest to the only albedo available in RACMO, which is calculated by dividing the outgoing with the incoming solar radiation at the surface and includes both direct and diffuse radiation.

L482: The albedo has somewhat improved in summer in R24, resulting in more melt, but is often too low. Note, however, that for R24 both direct and diffuse radiation are used to determine the albedo. This case study shows the importance of the albedo feedback on melt, and in particular the refreezing grain size that is a key parameter in modeling the snow melt-albedo feedback.

Fig 12: Time series of the broadband albedo of R24 and R23p2, which includes contributions of both direct and diffuse radiation, for the same grid points in the Amery ice shelf (upper panel) and King Baudouin ice shelf (lower panel).

**Technical corrections**

L. 11 "shortwave radiative fluxes": Net or downward or upward? Please specify.

L11: Temperature, upward and downward shortwave radiative fluxes and albedo are modeled well compared to in-situ observations.

L. 12 "Longwave radiative fluxes": Same as the technical comment on L. 11

L12: The longwave downward radiative and turbulent fluxes, however, require further model developments.

L. 32 "In Antarctica": This can be removed because it is already said in L. 30.

Done

L. 58 ~ 60: Almost the same information is introduced at the beginning of Sect. 2.1. I think this part in the introduction section can be removed.

Done, and changed the following:

L58: Recently, a new model version, RACMO2.4p1 (R24, Van Dalum et al., 2024b), has been developed and includes an update of the European Centre for Medium-Range Weather Forecast (ECMWF) Integrated Forecast System (IFS) physics, which is embedded in RACMO, from cycle 33r1 to cycle 47r1 (ECMWF, 2020).

Figure 2 caption and L. 213 "Yearly-average SMB": I feel this is a bit confusing. At first, I felt it referred to the value of monthly (or daily?) accumulated SMB averaged over a year. But maybe it's "average annual (accumulated) SMB"? If so, for the Figure 2 caption, I would write like "Annual accumulated SMB of R24 (a) averaged for 1979-2023 ~". If the authors agree with this point, please check other parts of this paper, e.g., Fig. 3 caption.

We agree and we changed it throughout the manuscript.

Table 1 caption: The difference should be introduced together with Δ before the standard deviation.

Done, now also introduced Δ in Table 2.

L. 229 "Fig. 2c with d": "Fig. 2c with Fig. 2d"?

Yes we meant Fig. 2c with Fig. 2d. Changed accordingly.

Table 2 caption: I think it is better to rephrase "~AntSMB observations (1979-2017) with respect to (w.r.t.) R24 and R23p2" -> "~ R24 and R23p2 with respect to (w.r.t.) AntSMB observations (1979-2017)". Please also check the header information in the table.

Done

L. 298: Suggest adding "in R24" at the end of "The interior of the EAIS thus seems to be too dry."

Changed the following:

L297: For B-C that also covers a considerable part of the East Antarctic Plateau, the SMB is now underestimated. The interior of the EAIS thus seems to be too dry in R24.

L. 318: "Radiative transfer" -> "Radiative transfer in snow and ice"?

Yes, changed accordingly.

Figure 9b: The intervals in the x and y axes can be changed from 50 to 100. The current presentation of x-axis is not good.

Done, also changed Fig. A2.

Figure 10b: The interval in the color bar should be improved. At present, there are three "0.00", two "0.01", and two "-0.01".

This is indeed confusing. We have changed the colorbar to the following, using scientific notation:

[Figure]

Figure 12a: Suggest adding "Modeled" before "Melt with respect to the QSCAT melt product regridded to ~"

Done

L. 513: It is helpful for readers to indicate the horizontal resolution of R23p2 again here. It is also informative if the authors list some technical updates achieved in R24 with respect to R23p2 here.

Following your suggestions, we updated the first paragraph (after the bullet points that are now part of the conclusion, as is requested by Review #2) of the conclusions to the following:

L510: For the near-surface climate and SMB product presented in this study, R24 is run on an 11 km horizontal resolution grid that includes Antarctica, the Drake Passage and the southern tip of South America for 1960-2023. We evaluated the model results with in-situ and remote sensing observations, and compared it to the previous operational RACMO version, RACMO2.3p2, which is run on a 27 km horizontal resolution grid for 1979-2023. With respect to RACMO2.3p2, several aspects of the model are updated in R24, including an update to the IFS physics module, a new multi-layer snow scheme for seasonal snow, an updated fractional ice mask and a new albedo and radiative transfer in snow scheme.

**Review #2 by Josep Bonsoms**

**General Comments**

The manuscript by van Dalum et al. presents a comprehensive evaluation of the new RACMO2.4p1 regional atmospheric climate model applied to the Antarctic Ice Sheet over the 1960–2023 period.

The results demonstrate improvements over the previous RACMO2.3p2 version, particularly in simulating key components of the SMB and near-surface climate. The integration of updated physical parameterizations—such as a revised blowing snow scheme or a spectral snow albedo model represent a step forward in polar regional modelling. Further, the manuscript provides robust validation using a multi-source approach (AntSMB, AntAWS, IMAU AWS, MODIS, QSCAT), while also acknowledging existing limitations, such as sublimation-related uncertainties in the ablation area.

This work fits well within the scope of the journal, provides valuable climate data for the scientific community, and represents an important contribution to the field. I recommend minor revisions to further improve clarity and interpretability.

There are a few specific areas where additional clarification or discussion could be considered to enhance the manuscript:

1. Sublimation and longwave radiation biases: The manuscript acknowledges the underestimation of downward longwave radiation (~20 W/m²), dry biases in the AIS interior and uncertainties related to sublimation, particularly in marginal zones. A more detailed discussion of these aspects would improve the understanding of model limitations. If available, consider including sensitivity tests or comparisons with alternative physical parameterizations to evaluate these biases more thoroughly.

Thank you for the suggestion, we have expanded the analyses of the uncertainties and have discussed how the model can be improved in more detail in the following way:

L255: Gadde and Van de Berg (2024) report on average a sublimation of blowing snow of 175 Gt yr-1 in their development version of RACMO2.3p3, which is higher than the calculated 101 Gt yr-1 in the regional climate model CRYOWRF. Gadde and Van de Berg (2024) employed RACMO2.3p3 on a horizontal resolution grid of 27 km and reported that compared to satellite observations of Palm et al. (2017), underestimation of blowing snow is possible near the coast as the resolution is not high enough to capture the large spatial gradients. The horizontal resolution of R24, however, is 11 km. In addition, Gadde and Van de Berg (2024) report a blowing snow erosion of 8 Gt yr-1, which is lower than 11.2 Gt yr-1 in R24, and the larger integrated sublimation of blowing snow is therefore expected.

L384: When considering only clear-sky observations, the bias is -25.4 W m-2. MARv3.11 shows a similar bias as R24, with a mean bias of -20.4 W m-2 without drifting snow and -14.9 W m-2 with drifting snow with respect to observations attached to a meteorological mast close to the coast of Adelie Land in east Antarctica (Le Toumelin et al., 2021).

L398: In future work, clouds and the impact on longwave radiation will be investigated by using the recently-launched EarthCARE satellite (Wehr et al., 2023), which may help identify

mismatches in cloud location, structure, height, composition or optical properties. For clear-sky conditions, R24 determines longwave radiation by calculating absorption of all major gases, such as water vapour, O2 and CO2 on sixteen spectral bands for each vertical layer using the layer's transmittance and temperature (Mlawer et al., 1997). Modeling gas and aerosol concentrations correctly are thus essential. Future work will therefore involve investigating the impact of CAMS aerosols and SOCRATES cloud optical properties, which are are embedded in R24, on longwave radiation and how well they are tuned for the polar regions. For turbulence, a turbulent kinetic energy (TKE) closure scheme can potentially improve modeled turbulent fluxes, as the introduction of TKE as a prognostic variable that can move by advection can be used to simulate eddy diffusivity, allowing non-local effects to impact turbulence (Lenderink and Holtslag, 2000, 2004).

2. Early-period output (1960–1979): As noted by the authors, outputs from the early period are less reliable due to limited satellite data availability. It could be interesting to considering the exclusion of this timeframe from mean statistics, or including uncertainty estimates to better reflect its lower reliability.

The early period of 1960-1979 is indeed less reliable than later periods due to limitations in satellite data availability. We have already excluded the 1960-1979 period from the statistics that are presented in our study, including the statistics shown in Table 1, 2 and 3, and from almost all analyses. Only in Fig. 4 and Fig. 13 is the period of 1960-1979 included. For Fig. 4 this is done to show and discuss the clear difference for the SMB before and after 1979 and that this difference originates mostly from East Antarctica. For Fig. 13, this early period is only used as a starting point (with the year 1962) to illustrate how the snow depth and snow cover gradually evolves over time on ice-free tiles. Omitting 1962 from this analysis would not alter the discussion and interpretation of Fig. 13. We have added the following to help clarify that 1960-1979 is not included in the statistics of AntAWS and AntSMB.

L270: With respect to the majority of in situ observations used in this study between 1979-2017 of the AntSMB data set, which cover the entire AIS but mostly originate from radar observations performed on the WAIS (145,427 observations)…

L291: (Table 2, where 1960-1979 are excluded from the statistics).

L380: Compared to the IMAU AWS observations between 1995 and 2022 …

L429: R24 compares well with near-surface AntAWS observations covering 1980-2021 (Wang et al., 2023) ...

3. Comparison with other models: While the comparison with RACMO2.3p2 is well presented, a brief discussion of how RACMO2.4p1 compares with other regional climate model outputs over the AIS (e.g., MAR or CRYOWRF) would provide additional context, even if a direct evaluation was not conducted. In addition, It should be stated whether the AWS data used for validation are assimilated into ERA5, as this could affect the independence of the evaluation.

A full model intercomparison between RACMO2.4p1 and e.g., MAR or CRYOWRF would be beyond the scope of this study. We have, however, now included a brief discussion at several places in the paper. We are also aware of and contributing to several ongoing studies where RACMO, MAR and other RCMs are compared, where a model intercomparison will be

covered in more detail. Furthermore, some AntAWS are assimilated in ERA5, but as RACMO calculates its own climate, a comparison is still appropriate. IMAU AWS are not assimilated into ERA5, which we now also mention in the paper.

L164: Some AntAWS observations are assimilated into ERA5, but a comparison with RACMO is appropriate, as RACMO calculates its own climate.

L171: Observations are not assimilated into ERA5 and are daily averaged…

L234: The introduction of advection of hydrometeors brings the modeled precipitation patterns qualitatively closer to that of the MAR regional climate model (Fig. 4 of Fettweis et al., 2020), where advection of hydrometeors is already calculated since several model versions.

L255 Gadde and Van de Berg (2024) reported on average a sublimation of blowing snow of 175 Gt yr-1 in their development version of RACMO2.3p3, which is higher than the calculated 101 Gt yr-1 in the regional climate model CRYOWRF.

L268: The integrated SMB of R24 is comparable to that of other regional climate models. Mottram et al. (2021) show that MARv3.10 and HIRHAM5 0.11 degrees model a higher integrated SMB for the AIS including ice shelves of 2633 and 2657 Gt yr-1, respectively, for the period 1980 to 2010. MetUM and COSMO-CLM, however, model lower SMB values, with 2327 and 2023 Gt yr-1. Mottram et al. (2021) note that differences in horizontal resolution are the main cause of intermodel SMB differences, in particular in the AP and WAIS, where steep slopes cause orographic precipitation. Qualitatively comparing Fig. 2a to the ensemble mean of Fig. 6a of Mottram et al. (2021) shows that the SMB of R24 is also spatially in good agreement, and has improved in particular on the coast of the EAIS compared to RACMO2.3 (Fig. 6e of Mottram et al. (2021)).

L384: When considering only clear-sky observations, the bias is -25.4 W m-2. MARv3.11 shows a similar bias as R24, with a mean bias of -20.4 W m-2 without drifting snow and -14.9 W m-2 with drifting snow with respect to observations attached to a meteorological mast close to the coast of Adelie Land in east Antarctica (Le Toumelin et al., 2021).

**Specific comments**

- L28: Add a citation to support the statement "…which is the root cause of recent net mass losses of the AIS."

  …which is the root cause of recent net mass losses of the AIS (Otosaka et al., 2023).

- L32: Add a reference for "…In Antarctica, ablation is typically the result of sublimation or drifting snow erosion."

  Ablation is typically the result of sublimation or drifting snow erosion (Richter et al., 2021, Gadde and Van de Berg, 2024).

- L67 (End of Introduction): Consider deleting this paragraph, as it appears unnecessary.

We normally include a short part at the end of the introduction to briefly give an overview of the main parts of the paper and in what section they can be found. This paragraph is such an overview, and we think that it is useful for the readers to have a quick overview about the remaining sections and to be able to quickly navigate to the specific parts.

- L105: "This results, for example, in a more realistic representation of mixed-phase clouds…" Consider splitting this sentence or rephrasing for better readability.

  We have changed it to:

  L105: Several processes are therefore represented more realistically, such as mixed-phase clouds.

- Figure 1: I suggest to add a legend showing measurement locations (e.g., AntSMB, IMAU AWS in red squares). This would allow removal of related description in the text (e.g., L157, L164) and improve figure clarity.

  We have added a legend and removed some text related to the description on line 157, 164, 171 and 430 and in the caption of Fig. 11

- Figure 2a: The 0 value is currently shown in blue. Why not changing to white for consistency with other figures (e.g., white = zero, blue = negative, yellow/red = positive)?

  We chose this color scheme and not a blue/white/red color scheme for the SMB in Antarctica due the very limited amount of areas where the SMB becomes negative. When a blue/white/red color scheme was chosen, almost the entire figure would be red, making it difficult to illustrate the SMB gradient from the interior of the EAIS towards the coast, which is clearly represented with the current color scheme. Furthermore, the current color scheme also distinctly shows the much higher SMB in the WAIS and the Antarctic Peninsula.

- L215–L219: This is interesting, however, consider adding regional averages or specific values with standard deviation, rather than descriptive text. This approach, done in Section 3.2, improves the quantitative interpretation.

  You are right and we have added the following to address this:

  Figure 2a shows the annual accumulated SMB of R24. Virtually all of the AIS ice sheet is characterized by a positive SMB, with particularly high values on the coast of West Antarctica and the western side of the AP (Fig. 5b and Fig. 5c, up to 2000 mm w.e. yr-1). The eastern side of the AP, which is separated climatologically from the western side by a narrow mountain range, is characterized by a considerably lower SMB, in particular on the Larsen C ice shelf (often lower than 500 mm w.e. yr-1). The coast of East Antarctica also has a high SMB in general, while the interior is dry and the SMB is much lower (Fig. 5d, often lower than 30 mm w.e. yr-1).

- Figure 4: The legend in panel (b) overlaps important details during a period of high-SMB uncertainty. I suggest moving it outside the plot in the right side. Also, consider replace symbols in figure legends with line's instead of points.

  You are right that the legend overlaps with some details, and we have moved it out to the right side of the figure. In the legend we chose to represent the SMB components with a symbol and not a line such that there is no confusion that the colors are used for both RACMO2.4 (solid line) and RACMO2.3p2 (dashed lined). If a line would be used for the colors, it may suggest that the colors are only valid for RACMO2.4, which is not the case. Hence only two lines are shown in black to illustrate the two line styles used for the models, while the colors are shown with a symbol.

- L280–L290: Clarify how regionalization of the AIS is done in the methods sections, If it is based on IMBIE conventions, I suggest including this detail in the Methods section and not repeat it hereafter.

  It is indeed based on the IMBIE conventions, as is mentioned on line 290. As requested, we have moved the mention of the IMBIE convention from line 290 to line 157.

  L290: Splitting the AIS in drainage basins reveals that the SMB has changed considerably for several basins (Table 2).

  L157: Locations of in situ SMB measurements used in this study are shown as black dots in Fig. 1 and will also be categorized by drainage basin following the Ice Sheet Mass Balance Inter-comparison Exercise (IMBIE) convention, which is based on Rignot et al. (2019).

- L451: In the sentence starting "At the end of December…", please specify exact dates.

  Done

  L451: On the 25th of December, melt decreases in R23p2 and virtually no melt is modeled from the 2nd of January onward with the arrival of fresh snow. As no radiative transfer is calculated in R23p2, the albedo increases strongly even with a thin snowpack, inhibiting melt. In R24, however, melting continues until the fresh snow layers grow thick enough on the 28th of January.

  L461: A similar signal is found for the King Baudouin ice shelf (Fig. 12c), where melting starts on the 24th of November and persists through January in R24, which is in agreement with AMSR-E liquid water observations (lower panel of Fig. 12c), while almost no melting occurs in January in R23p2. Apart from roughly a week with more cloud cover in mid January in R23p2 (from the 16th to the 24th of January), the SEB components of R24 are also similar to R23p2 in January,

**Conclusion**

The paper may be extensive for readers who are not familiar with the topic. I suggest including a brief list with bullet points of the main findings in the conclusion to help readers recap before the final remarks.

We have added the following bullet points and changed the first sentence of the first paragraph to help readers recap before the final remarks:

L510:

- We presented a new near-surface climate and SMB product for the Antarctic ice sheet using the new model version RACMO2.4p1 (R24).
- Integrated over the ice sheet, including ice shelves, the modeled SMB is 2546 Gt yr-1, with an interannual variability of 133 Gt yr-1.
- Compared to RACMO2.3p2, the SMB in the WAIS and the AP have improved due to precipitation and sublimation changes caused by the advection of snow hydrometeors and alterations in the blowing snow parameterization.
- Melt events are modeled around the AIS and compare well with remote sensing observations.
- Compared to AWS observations, shortwave radiative fluxes, albedo and near-surface temperature are modeled well, but further research is needed to improve downward longwave radiation and turbulence.
- Two case studies are presented that show the impact of albedo and radiative transfer on melt and explore the impact of the introduction of a fractional glacier mask.

For the near-surface climate and SMB product presented in this study, R24 is run on an 11 km horizontal resolution grid that includes Antarctica, the Drake Passage and the southern tip of South America for 1960-2023.

---

## Author Response (AR2)

Referee comment responses on the manuscript:

**The surface mass balance and near-surface climate of the Antarctic ice sheet in RACMO2.4p1**

by C.T. van Dalum et al.

We would like to thank the referee to take a final look at our paper and provide a comment. In orange the response, in blue the changes that we would implement in the manuscript. All line numbers refer to the last manuscript version.

**Review**

General comments:

I would like to thank the authors for considering my earlier comments. I have found that most of the authors' responses to my earlier concerns are convincing. I suggest that this paper can be published once the authors address the following minor issue.

Regarding the MODIS white-sky snow albedo shown in Fig. 12, the authors' response indicates that "We deemed the white-sky MODIS albedo product to be the closest to what we have available in RACMO." Related to this point, in the revised manuscript, it is indicated that "The MODIS white-sky albedo is closest to the only albedo available in RACMO, which is calculated by dividing the outgoing with the incoming solar radiation at the surface and includes both direct and diffuse radiation." (L. 192 ~ 194, manuscript v2) However, I would like to note that the description is insufficient because some readers may still struggle to understand the differences between the RACMO albedo and the MODIS white-sky albedo. I think the authors should explain more specifically that the broadband snow albedo under overcast conditions, when the most direct beam disappears and the snow surface is illuminated diffusely, loses solar zenith angle dependence and the snow albedo value becomes higher than those at any solar zenith angle for a clear-sky condition (Liljequist, 1956; Aoki et al., 1999), which implies that the MODIS white-sky albedo shown in Fig. 12d can be higher than the actual snow albedo.

References:

Aoki, T., Aoki, T., Fukabori, M., and Uchiyama, A.: Numerical simulation of the atmospheric effects on snow albedo with multiple scattering radiative transfer model for the atmosphere-snow system, J. Meteorol. Soc. Jpn., 77, 595–614, 1999.

Liljequist, G. H.: Energy Exchanges of an Antarctic Snow-Field: Short-Wave Radiation, Norwegian-British-Swedish Antarctic Expedition (Maudheim, 71°03' S, 10°56' W), 1949–52, Scientific Results, Vol. 2, Part 1A, Norsk Polarinstitutt, Oslo, 107 pp., 1956.

Thank you for your comment. You are right that some more explanation is appropriate. Based on your comment, we have changed L192-L194 and added the following in the manuscript (based on manuscript v2):

L192: Under overcast conditions, radiation loses its solar zenith angle dependence, as the direct radiation component is lost and the snow surface is illuminated diffusely. In addition to a spectral shift towards visible light that typically occurs for cloudy conditions, for which the spectral albedo is high (Dang et al., 2015), the resulting broadband albedo increases with cloud cover and is higher than those at any solar zenith angles for clear-sky conditions (Liljequist, 1956; Aoki et al., 1999). These effects are not included in the MODIS white-sky albedo product, and some discrepancies are therefore expected when comparing with the only albedo available in RACMO, which is calculated by dividing the outgoing with the incoming solar radiation at the surface and includes both direct and diffuse radiation.

L525: Note, however, that for RACMO both direct and diffuse radiation are used to calculate the albedo, while MODIS white-sky albedo is determined in the absence of a direct component. This case study shows the importance of the albedo feedback on melt, and in particular the refreezing grain size that is a key parameter in modeling the snow melt-albedo feedback.